**Special Section:**
The COVID-19 pandemic: linking health, society and environment

**Key Points:**
- 2020 is the only year that both Europe and western North America show strong negative tropospheric ozone anomalies since 1994
- Positive free tropospheric ozone trends above Europe and western North America since 1994 are diminished by the 2020 anomalies
- Data integration of multiple time series provides a better understanding of ozone variability compared to individual records

**Supporting Information:**
Supporting Information may be found in the online version of this article.

**Correspondence to:**
K.-L. Chang,
kai-lan.chang@noaa.gov

**Citation:**
Chang, K.-L., Cooper, O. R., Gaudel, A., Allaart, M., Ancellet, G., Clark, H., et al. (2022). Impact of the COVID-19 economic downturn on tropospheric ozone trends: An uncertainty weighted data synthesis for quantifying regional anomalies above western North America and Europe. *AGU Advances*, *3*, e2021AV000542. https://doi.org/10.1029/2021AV000542

**Peer Review** The peer review history for this article is available as a PDF in the Supporting Information.

# Impact of the COVID-19 Economic Downturn on Tropospheric Ozone Trends: An Uncertainty Weighted Data Synthesis for Quantifying Regional Anomalies Above Western North America and Europe

Kai-Lan Chang[1,2] , Owen R. Cooper[1,2] , Audrey Gaudel[1,2] , Marc Allaart[3], Gerard Ancellet[4] , Hannah Clark[5], Sophie Godin-Beekmann[4] , Thierry Leblanc[6] , Roeland Van Malderen[7] , Philippe Nédélec[8] , Irina Petropavlovskikh[1,9] , Wolfgang Steinbrecht[10] , René Stübi[11] , David W. Tarasick[12] , and Carlos Torres[13]

[1]Cooperative Institute for Research in Environmental Sciences, University of Colorado, Boulder, CO, USA, [2]NOAA Chemical Sciences Laboratory, Boulder, CO, USA, [3]Royal Netherlands Meteorological Institute, De Bilt, The Netherlands, [4]LATMOS, Sorbonne Université-UVSQ-CNRS/INSU, Paris, France, [5]IAGOS-AISBL, Brussels, Belgium, [6]Jet Propulsion Laboratory, California Institute of Technology, Wrightwood, CA, USA, [7]Royal Meteorological Institute of Belgium, Uccle, Belgium, [8]Laboratoire d'Aérologie, CNRS and Université de Toulouse III, Toulouse, France, [9]NOAA Global Monitoring Laboratory, Boulder, CO, USA, [10]Deutscher Wetterdienst, Hohenpeissenberg, Germany, [11]Federal Office of Meteorology and Climatology, MeteoSwiss, Payerne, Switzerland, [12]Environment and Climate Change Canada, Toronto, ON, Canada, [13]Izaña Atmospheric Research Center, AEMET, Tenerife, Spain

**Abstract** This study quantifies the association between the COVID-19 economic downturn and 2020 tropospheric ozone anomalies above Europe and western North America, and their impact on long-term trends. Anomaly detection for an atmospheric time series is usually carried out by identifying potentially aberrant data points relative to climatological values. However, detecting ozone anomalies from sparsely sampled ozonesonde profiles (once per week at most sites) is challenging due to ozone's high temporal variability. We first demonstrate the challenges for summarizing regional trends based on independent time series from multiple nearby ozone profiling stations. We then propose a novel regional-scale anomaly detection framework based on generalized additive mixed models, which accounts for the sampling frequency and inherent data uncertainty associated with each vertical profile data set, measured by ozonesondes, lidar or commercial aircraft. This method produces a long-term monthly time series with high vertical resolution that reports ozone anomalies from the surface to the middle-stratosphere under a unified framework, which can be used to quantify the regional-scale ozone anomalies during the COVID-19 economic downturn. By incorporating extensive commercial aircraft data and frequently sampled ozonesonde profiles above Europe, we show that the complex interannual variability of ozone can be adequately captured by our modeling approach. The results show that free tropospheric ozone negative anomalies in 2020 are the most profound since the benchmark year of 1994 for both Europe and western North America, and positive trends over 1994–2019 are diminished in both regions by the 2020 anomalies.

**Plain Language Summary** Ozone in the free troposphere has increased across the Northern Hemisphere since the mid-1990s. However, the decrease of ozone precursor emissions due to the COVID-19 economic downturn likely explains the unusual free tropospheric negative ozone anomalies previously observed during spring and summer of 2020. This work investigates ozone trends and anomalies in the free troposphere using a new regional scale method for merging separate ozone time series, with focus on anomalies during the full 12 months of 2020. We found that the positive 1994–2019 ozone trends above Europe and western North America are diminished when including the large negative anomalies in 2020, and 2020 is the only year in which both regions show coincident and profound negative anomalies since the benchmark year of 1994.

## 1. Introduction

Tropospheric ozone is a greenhouse gas that contributes to climate change, and an air pollutant detrimental to human health and agriculture (Fleming et al., 2018; Gaudel et al., 2018; Mills et al., 2018). The tropospheric ozone burden is largely maintained by photochemical ozone production involving precursor gases, such as nitrogen

**Author Contributions:**
**Conceptualization:** Kai-Lan Chang, Owen R. Cooper
**Data curation:** Audrey Gaudel, Marc Allaart, Gerard Ancellet, Hannah Clark, Sophie Godin-Beekmann, Thierry Leblanc, Roeland Van Malderen, Philippe Nédélec, Irina Petropavlovskikh, Wolfgang Steinbrecht, René Stübi, David W. Tarasick, Carlos Torres
**Formal analysis:** Kai-Lan Chang
**Investigation:** Kai-Lan Chang, Owen R. Cooper
**Methodology:** Kai-Lan Chang
**Software:** Kai-Lan Chang
**Supervision:** Owen R. Cooper
**Validation:** Kai-Lan Chang
**Visualization:** Kai-Lan Chang
**Writing – original draft:** Kai-Lan Chang, Owen R. Cooper, Audrey Gaudel
**Writing – review & editing:** Kai-Lan Chang, Owen R. Cooper, Hannah Clark, Roeland Van Malderen, Irina Petropavlovskikh, David W. Tarasick, Carlos Torres

oxides (NOx), methane, carbon monoxide and volatile organic compounds (VOC; Archibald et al., 2019; Young et al., 2018). Chemistry-climate models indicate that the global tropospheric ozone burden has increased since 1850 (Griffiths et al., 2021; Skeie et al., 2020), and the available in situ observations show that this increase has continued in the free troposphere since the 1990s, especially in the tropics and the northern extratropics (Chang et al., 2020; Cohen et al., 2018; Cooper et al., 2020; Gaudel et al., 2018, 2020; Gulev et al., 2021; Liao et al., 2021; Tarasick et al., 2019).

In late 2019 a new coronavirus (SARS-CoV-2) emerged, causing the respiratory illness known as COVID-19. The resulting COVID-19 pandemic triggered a worldwide economic downturn in 2020 which reduced emissions of ozone precursor gases. These emissions reductions appeared to be related to a range of impacts on observed levels of ozone and particulate matter at the surface and in the free troposphere (Bauwens et al., 2020; Clark et al., 2021; Cooper et al., 2021; Cristofanelli et al., 2021; Gkatzelis et al., 2021; Keller et al., 2021; Kondragunta et al., 2021; Le et al., 2020; Liu et al., 2020; Shi & Brasseur, 2020; Sokhi et al., 2021; Steinbrecht et al., 2021). In general, surface observations from monitoring networks in many nations showed that urban areas with NOx-limited regimes (e.g., Rio de Janeiro and South African urban areas) experienced ozone decreases, while urban areas with VOC-limited regimes experienced ozone increases (e.g., urban areas in South Korea and Colombia; Sokhi et al., 2021). Several new attribution studies have estimated the global scale response of tropospheric ozone to the emissions reductions in 2020 (Bouarar et al., 2021; Gaubert et al., 2021; Miyazaki et al., 2021; Weber et al., 2020). These model-based studies concluded that some regions experienced surface ozone decreases due to reduced photochemical production, while many urban areas experienced ozone increases due to lower NO emissions, which limited the ozone destruction that typically occurs in highly polluted urban centers (Sillman, 1999). In terms of the free troposphere, the model analysis by Miyazaki et al. (2021) indicated that the global tropospheric ozone burden decreased by 2%, and mid-troposphere ozone above northern mid-latitudes was reduced by 2–3 ppb, while Bouarar et al. (2021) calculated an ozone decrease of 4%–8% above the northern extratropics due to spring and summer emissions reductions. These modeled ozone reductions are consistent with the observed ozone decreases of approximately 7% at multiple ozone profile monitoring locations across the northern extratropics, focusing on the 1–8 km column and April-August 2020 (Steinbrecht et al., 2021).

The goal of our analysis is to extend the initial findings of Steinbrecht et al. (2021) to produce a more detailed understanding of the 2020 free tropospheric ozone anomalies above Europe and western North America, as follows: (a) Steinbrecht et al. (2021) focused on a single integrated tropospheric column (1–8 km above sea level) over April-August 2020; here we investigate the detailed ozone variations throughout the depth of the troposphere for all 12 months of the year; (b) the previous findings were based on a simple average of observations from all available ozone profiling stations across the Northern Hemisphere, with no adjustments to account for sampling biases; here we focus on Europe and western North America, the two regions with the densest monitoring, using a new statistical method to determine regional-scale anomalies; (c) the previous effort compared the 2020 ozone anomalies to a fixed 2000–2019 baseline period, while our analysis calculates long-term trends (1994–2019) and quantifies the impact of the 2020 anomalies on these trends; and (d) our analysis includes the extensive 1994–2020 IAGOS (In-Service Aircraft for a Global Observing System) commercial aircraft ozone record, which triples the number of ozone profiles available from the ozonesonde sites above central Europe (albeit IAGOS profiles are less frequent in 2020 compared to previous years).

Importantly, to quantify regional-scale anomalies, we developed a novel two-step regional-scale anomaly detection framework based on generalized additive mixed models (GAMM; Wood, 2006), which accounts for the sampling frequency and inherent data uncertainty associated with each vertical profile data set, measured by ozonesondes, lidar or commercial aircraft. This method produces a fused long-term monthly time series with high vertical resolution that reports ozone anomalies from the surface to the middle-stratosphere, which can be used to quantify the regional-scale ozone anomalies under a consistent and unified framework. Based on ~45,700 ozone profiles above Europe and ~9,900 profiles above western North America, the result is a refined estimate of the regional-scale ozone anomalies above Europe and western North America during the COVID-19 economic downturn, and we also quantify the impact of the 2020 ozone anomalies on the long-term trends (1994–2020). A clear advantage of this regional-scale approach to quantifying ozone trends is that it incorporates data from multiple sources for the identification of systematic variations; in contrast, trend detection at individual sites can be a challenge due to low signal-to-noise ratios (Chang et al., 2020; Saunois et al., 2012).

Our two-step regional-scale anomaly detection framework was carefully designed to distinguish meaningful anomalies from outliers. The formal definition of an outlier can be recognized as a data point that shows a substantial deviation from the other data points in a sample or a time series, therefore it is reasonable to suspect that this data point is generated by a different process (Aggarwal, 2015; Hawkins, 1980); with respect to tropospheric ozone the processes in question are either photochemical production/destruction or unique meteorological conditions that introduce air masses that are highly enriched or highly depleted in ozone. However, anomalies are not limited to outliers, and also include level shifts (Weatherhead et al., 2017), a series of constant values, or data that substantially deviate from usual variations or seasonality (though not as extreme as an outlier; Foorthuis, 2021). In this study, our anomaly detection method is not limited to the identification of outliers, but is designed to quantify changes in vertical ozone profiles that are potentially linked to the COVID-19 economic downturn.

In the first step of our two-step approach to quantifying regional ozone anomalies, we closely follow the statistical framework developed by Chang et al. (2020) for analyzing vertical profile data. This approach takes the vertical correlation structure in time series from neighboring pressure levels into account, yielding an improved quantification of ozone systematic change. For the anomaly quantification in time series of vertical ozone profiles, we found that it is most effective to investigate multiple vertical layers simultaneously, since we expect that meaningful anomalies are likely to be detectable across multiple neighboring pressure layers (though with different magnitudes), instead of being isolated to a single, narrow (e.g., 10 hPa) pressure layer.

However, the absolute magnitude of ozone variation in the upper troposphere can be very large with respect to the lower troposphere due to the higher ozone values and higher variability resulting from troposphere-stratosphere interactions. Therefore, it becomes difficult to discern an anomaly in the middle or lower troposphere, where ozone values appear to be steady when compared to the extreme levels of upper tropospheric and stratospheric ozone variability. We address this issue by making further adjustments to the framework developed by Chang et al. (2020), which then allows the technique to serve as a unified approach for coherent anomaly detection in the troposphere and stratosphere.

In the second step of our two-step approach to quantifying regional ozone anomalies, we apply a generalized additive mixed model (GAMM), which accounts for the sampling frequency and inherent data uncertainty associated with each vertical profile data set in a geographic region, such as Europe and western North America, as measured by ozonesondes, lidar or commercial aircraft. Our GAMM approach is an advancement over previous studies that calculated regional ozone trends from multiple profile measurement locations (Cooper et al., 2010; Gaudel et al., 2020), but employed less rigorous approaches for addressing spatial and temporal sampling variability across the study region.

As an introductory demonstration, Section 2 provides an update on observed ozone trends in the free troposphere through the end of 2020, based on individual ozone time series as measured by ozonesondes, lidar or commercial aircraft, highlighting the difficulty of summarizing the regional variations. Section 3 then provides the detailed description of our new statistical framework for improved regional anomaly quantification. Section 4 presents the quantified trends and anomalies above Europe and western North America. This study closes in Section 5 with a discussion and conclusions.

## 2. Ozone Trends Derived by Sparse Monitoring Stations

### 2.1. Ozone Profile Data

Ozone observations in the free troposphere are infrequently sampled and sparse in spatial coverage. Previous studies have recommended sampling frequencies of 12 (Saunois et al., 2012) to 20 profiles per month (Logan, 1999) for accurate quantification of monthly mean ozone values in the free troposphere, and 18 profiles per month for accurate trend detection using standard linear regression techniques (Chang et al., 2020). However, as shown below, very few monitoring locations achieve such sampling frequencies, highlighting the challenges for the detection of trends and anomalies in the free troposphere. Table 1 provides a list of monitoring locations available for this study (a map view of their locations is provided in Figure S1 in Supporting Information S1) and further details of the monitoring locations and programs are provided below; data access is described in the data availability section. It should be noted that although the operation at some stations can be extended back to the 1960s, the study period for the trend and anomaly detection is limited to 1994–2020 (some stations may be

**Table 1**
*List of Monitoring Stations*

| Site | Latitude | Longitude | Monitoring institution or network | Availability | #Profiles |
|---|---|---|---|---|---|
| N. America | | | | | |
| Edmonton, Canada | 53.55 | −114.10 | Meteorological Service of Canada | 11/70–12/20 | 2164 (1263) |
| Port Hardy, Canada | 50.43 | −127.29 | Meteorological Service of Canada | 06/18–12/20 | 102 |
| Kelowna, Canada | 49.53 | −119.29 | Meteorological Service of Canada | 11/03–06/17 | 700 |
| Trinidad Head (THD), California | 41.05 | −124.15 | NOAA Global Monitoring Laboratory | 08/97–12/20 | 1241 |
| Boulder, Colorado | 39.99 | −105.26 | NOAA Global Monitoring Laboratory | 04/67–12/20 | 1933 (1436) |
| Table Mountain (TMF), California | 34.40 | −117.70 | NASA Jet Propulsion Laboratory | 03/02–12/20 | 2862 |
| IAGOS | | | | 11/94–12/19 | 2293 |
| Europe | | | | | |
| Legionowo, Poland | 52.40 | 20.97 | Polish Institute of Meteorology and Water Management | 06/93–12/20 | 1458 (1431) |
| Lindenberg, Germany | 52.22 | 14.12 | Meteorological Observatory Lindenberg | 07/92–12/20 | 1224 (1132) |
| De Bilt, Netherlands | 52.10 | 5.18 | Royal Netherlands Meteorological Institute | 11/92–12/20 | 1484 (1434) |
| Uccle, Belgium | 50.80 | 4.36 | Royal Meteorological Institute of Belgium | 01/69–12/20 | 6799 (3913) |
| Hohenpeissenberg (HPB), Germany | 47.80 | 11.01 | Meteorological Observatory Hohenpeissenberg | 11/66–12/20 | 5897 (3422) |
| Payerne, Switzerland | 46.81 | 6.94 | MeteoSwiss | 09/68–12/20 | 6632 (4097) |
| Haute Provence (OHP), France | 43.92 | 5.71 | Haute-Provence Observatory | 01/91–12/20 | 1235 (1123) |
| Madrid, Spain | 40.45 | −3.72 | Spanish Meteorological Agency | 12/94–12/20 | 1104 |
| IAGOS | | | | 08/94–12/20 | 31727 |
| Oceania | | | | | |
| Broadmeadows, Australia | −37.69 | 144.95 | Australian Bureau of Meteorology | 02/99–12/20 | 1026 |
| Lauder, New Zealand | −45.04 | 169.68 | National Institute of Water and Atmospheric Research of New Zealand at Lauder | 08/86–12/20 | 1989 (1464) |
| Macquarie Island, Australia | −54.50 | 158.94 | Australian Bureau of Meteorology | 03/94–12/20 | 1127 |
| Other | | | | | |
| Tateno, Japan | 36.05 | 140.13 | Japan Meteorological Agency | 11/68–12/20 | 1890 (1373) |
| Izaña, Spain | 28.41 | −16.53 | Spanish Meteorological Agency | 01/95–12/20 | 1352 |
| Hong Kong, China | 22.31 | 114.17 | Hong Kong Observatory | 01/00–12/20 | 930 |
| Hilo, Hawaii | 19.72 | −155.07 | NOAA Global Monitoring Laboratory | 09/82–12/20 | 1749 (1336) |

*Note*. The number in parentheses indicates the total number of profiles after 1994

subject to intermittent data gaps and/or incomplete record), in order to incorporate the abundant measurements from the IAGOS program, which began in 1994.

1. Ozonesondes launched by NOAA's Global Monitoring Laboratory have a typical sampling frequency of once per week at Boulder (Colorado), Trinidad Head (THD, California) and Hilo (Hawaii) in recent decades. However, during infrequent field campaigns of limited duration the launch frequency was increased to several times per week or even daily, for example, Boulder had increased launches during the spring of 2020 to monitor the potential impact of the COVID-19 economic downturn
2. The lidar operated at Table Mountain Facility (TMF, California) produced ozone profiles with varying frequency (typically 2–4 profiles per week) from 2002 to 2018. In order to validate the satellite retrievals from the Tropospheric Monitoring Instrument (TROPOMI), TMF has operated the lidar during the daily TROPOMI overpasses since January 2018 (Chouza et al., 2019). Since TMF is a high elevation site (2285 m), we only consider the TMF data above 600 hPa in this study

3. The weekly ozonesonde operation at Kelowna, British Columbia started in November 2003 and ended in June 2017. To focus on baseline ozone at the west coast of Canada, the operation relocated to Port Hardy on the northern tip of Vancouver Island in June 2018. These sites are more than 300 km apart and the boundary layer processes that impact ozone along the North Pacific coastline (Port Hardy) and the interior of North America (Kelowna) clearly prevent us from calculating meaningful long-term trends in the boundary layer (below 3 km altitude). However, based on the long correlation distances between ozonesonde sites (Liu et al., 2009), these sites are close enough that they can be combined for the purposes of calculating trends in the free troposphere (above 3 km altitude). Figure S2 in Supporting Information S1 shows the 95th, 50th and fifth percentiles of the time series in the upper, middle and lower troposphere above Kelowna (2003–2017) and Port Hardy (2018–2020), respectively. In general, the medians and the magnitudes of variability are similar at different pressure layers, with greater 95th and lower fifth percentiles observed in the mid-troposphere above Port Hardy. However, this difference may be due to the short record at Port Hardy

4. Free tropospheric ozone is routinely monitored above Europe from several locations. Hohenpeissenberg (HPB, Germany), Payerne (Switzerland) and Uccle (Belgium) launch ozonesondes with a sampling frequency of two or three times a week. The other European sites used here (Legionowo, Poland; Lindenberg, Germany; De Bilt, Netherlands; Observatoire de Haute-Provence (OHP), France; Madrid, Spain) have a sampling frequency of approximately once a week

5. The IAGOS program is an important source of worldwide tropospheric ozone observations. Since 1994 IAGOS has measured ozone from multiple commercial aircraft using standard UV-absorption instruments, which are calibrated annually to a reference analyzer at the French Laboratoire National d'Essais, and compared to an in-flight ozone calibration source every 2 hr (Blot et al., 2021; Nédélec et al., 2015). IAGOS data have shown internal consistency for the duration of the program (Blot et al., 2021), have shown consistency with ozonesonde records in the upper troposphere-lower stratosphere above western Europe (Staufer et al., 2013, 2014), have been compared to regional surface and free tropospheric ozonesonde records (Logan et al., 2012; Petetin et al., 2018), and have been used to evaluate regional-scale ozone trends across the northern hemisphere (Cohen et al., 2018; Gaudel et al., 2020). Due to the known calibration history of the IAGOS instruments, the program's ozone observations can be considered a reference data set (Tarasick et al., 2019). The IAGOS data set contains over 31,000 profiles above central Western Europe (99% of profiles from Frankfurt, Paris, Munich, Brussels, Dusseldorf and Amsterdam) for the period 1994–2020, for an average sampling frequency of more than 100 profiles per month; above western North America, most profiles are sampled from Vancouver, San Francisco, Portland, and Seattle, with a total amount of ∼2300 profiles. Demonstrates the large number of IAGOS profiles above Europe with respect to ozonesonde records from sites such as HPB, Payerne, Uccle and OHP, where sample sizes at OHP represent the typical sampling frequency of once a week. The high sampling frequency has allowed previous studies to calculate ozone trends above Western Europe using the IAGOS data set alone (Cooper et al., 2020; Gaudel et al., 2020)

IAGOS data are more limited in 2020 due to the reduced aircraft operations during the COVID-19 pandemic, with some data available at Frankfurt airport. Clark et al. (2021) used these data to show that positive ozone anomalies near the surface were probably linked to decreased NOx, but that there was an important impact of exceptional meteorological conditions. In the free troposphere, they found that ozone levels were slightly lower than seen over the previous 26 years. In the current analysis we use IAGOS data for trend detection above Europe, and we also merge these profiles with those from nearby ozonesonde and lidar sites to produce continuous ozone records for anomaly detection above Western Europe and western North America.

1. While the focus of this study is on Europe and western North America we also explored the impact of ozone anomalies on trends at several other sites around the world to provide a broader global context. These sites include: Hilo (Hawaii), Tateno (Japan), Izaña (Spain), Hong Kong (China), Broadmeadows (Australia), Lauder (New Zealand) and Macquarie Island (Australia). The locations of these sites are shown in Figure S1 in Supporting Information S1

## 2.2. Updated Ozone Trends at Individual Sites

As mentioned above, Steinbrecht et al. (2021) found a 7% decrease in free tropospheric (1–8 km altitude) ozone across the Northern Hemisphere extratropics during spring and summer 2020, compared to the 2000–2020 climatological mean. Our expanded analysis, described below, shows that the 2020 free tropospheric anomalies

persisted through autumn and early winter of 2020. We begin with a relatively simple trend analysis method, to demonstrate the impacts of the COVID-19 economic downturn on ozone trends in the free troposphere (aggregated over 700-300 hPa), by comparing the 1994–2019 and 1994–2020 ozone trends for each of the stations listed in Table 1. These stations are mainly clustered in Europe and western North America, but we also include several stations from East Asia and the Southern Hemisphere, plus single sites in the central North Pacific Ocean (Hilo, Hawaii) and the eastern North Atlantic Ocean (Izaña). The trends and associated uncertainty are estimated based on the multiple linear regression model that we have used previously (Cooper et al., 2020), with additional terms for the quasi-biennial oscillation (QBO) and El Niño-Southern Oscillation (ENSO) in order to account for their potential correlations with ozone in the upper troposphere (Neu et al., 2014; Weatherhead et al., 2000; Ziemke et al., 1997):

$$y_t = \beta_0 + \beta_1 t + \beta_2 \text{ENSO} + \beta_3 \text{QBO}_{30} + \beta_4 \text{QBO}_{50} + \beta_5 \sin\left(2\pi\frac{\text{Month}}{12}\right) + \beta_6 \cos\left(2\pi\frac{\text{Month}}{12}\right) + N_t, \quad (1)$$

where $y_t$ is the ozone time series with a monthly temporal index $t$, $\beta_0$ is the intercept, $\beta_1$ is the linear trend estimate, $\beta_2$ is the coefficient associated with ENSO index, $\beta_3$ and $\beta_4$ are coefficients associated with the monthly mean zonal wind at 30 and 50 hpa (Liu et al., 2009; Soukharev, 1997), respectively (ENSO and QBO data links can be found in the data availability section), $\beta_5$ and $\beta_6$ are coefficients associated with harmonic functions representing jointly the seasonal cycle (which can be removed if data are deseasonalized in advance), and $N_t$ is the residual series. We use the AR (1) process to represent the remaining autocorrelation in the residuals. We also calculate the relative change of trends in 2020 with respect to the trends through 2019, that is, let $b$ and $b'$ be the trend estimate through the end of 2019 and 2020; the relative change in percentage is calculated by $100 \times (b' - b)/|b|$.

The free tropospheric (700-300 hPa) ozone trend estimate and associated 2-sigma uncertainty for each station are provided in Table 2. We summarize the key findings as follows:

1. Above western North America, Boulder and Edmonton show a clear decrease, and IAGOS shows a clear increase from 1994 to 2019, while the remaining sites show a range of weak positive or weak negative trends. When the year 2020 is included, the trends at THD become more strongly negative, but the trends at the other sites remain weak
2. Above Europe trends through 2019 are positive for IAGOS, De Bilt, OHP and Uccle, negative for Legionowo, Lindenberg and Payerne, with little change at HPB and Madrid. The inclusion of 2020 forces the positive trends to diminish and the negative trends to lean even more negative
3. Above the Oceania sites (Broadmeadows, Lauder and Macquarie Island) in the Southern Hemisphere the trends through 2019 change little when the year 2020 is included
4. Above the remaining Northern Hemisphere sites, the trends through 2019 are mixed, but all shift toward the negative end of the spectrum when 2020 is included (Tateno, Izaña and Hilo), except for the tropical site of Hong Kong where little change was found

The trend estimates and associated statistics for various tropospheric layers are reported in Supplementary Table 1. While we were able to easily investigate the impact of the COVID-19 economic downturn on free tropospheric ozone trends above individual sites, it remains unclear how to summarize the regional variations, especially when a wide range of changes was found at the European sites. For example, (a) the large change in trends at HPB and Madrid when 2020 data are included is mainly due to the relatively small magnitude of trends through 2019; and (b) the trend above Lindenberg should not be over-interpreted due to a large data gap before the COVID-19 period. Another major discrepancy concerns the variability in long-term trends, with both Europe and western North America having a range of positive and negative trends through 2019.

The high degree of variability in the magnitude and uncertainty of trends within Europe and western North America is not unexpected given ozone's high temporal and spatial variability and the low sampling frequency of the free troposphere. Our previous work demonstrated that a sampling frequency of 18 profiles per month is necessary for accurate trend quantification in the free troposphere, yet the ozonesonde sites are limited to much lower sampling frequencies of 4–12 profiles per month. Studies that have relied on the high sampling frequency of the IAGOS program (Cohen et al., 2018; Gaudel et al., 2020), or have combined IAGOS data with ozonesonde profiles (Cooper et al., 2010), have quantified Northern Hemisphere regional-scale ozone trends that are generally positive, and consistent with satellite and model studies of Northern Hemisphere ozone trends, as recently

assessed by the Intergovernmental Panel on Climate Change (Gulev et al., 2021). Based on this approach of combining all available observations to explore regional-scale trends, we now propose a new statistical methodology to integrate multiple observational records to produce an estimate of regional-scale variations.

Our previous work has shown that free tropospheric ozone has a high degree of spatial and temporal variability across mid-latitude, western North America on daily, weekly and monthly time scales (Cooper et al., 2007; Cooper et al., 2011; Lin, Fiore, Cooper, et al., 2012; Lin, Fiore, Horowitz, et al., 2012), with much greater variability in the boundary layer and upper troposphere than in the mid-troposphere (the focus of the current study). When focusing on multi-year time periods (≥5 years) the average mid-tropospheric ozone distribution across mid-latitude North America reveals only small gradients in ozone mixing ratios (Cooper et al., 2010, 2011; Gaudel et al., 2018), and satellite-detected trends of tropospheric column ozone are similar across this region (Ziemke et al., 2019). These observations are consistent with the analysis of Liu et al. (2009) which determined that free tropospheric ozone observations are spatially correlated over distances of 500–1000 km. Based on these previous studies, this analysis weights all mid-tropospheric ozone observations above western North America equally in terms of their spatial representativeness, and instead focuses on the temporal variability on monthly and yearly time scales to quantify the long-term, regional ozone trend. A limitation of this approach is that the trend estimate is assumed to be representative of the entire region, and we do not have enough data to explore the spatial variability of trends across the region. We apply the same methodology to the much smaller western Europe study region (one eighth the size of the western North America region), which also shows limited gradients in mid-tropospheric ozone distribution and trends when focusing on multi-year time periods (≥5 years; Gaudel et al., 2018; Ziemke et al., 2019).

## 3. Statistical Method

To introduce our statistical framework, we first review the concept of mixed modeling for integrating a variety of sources of (potentially heterogeneous) data (Section 3.1). We then discuss the statistical principles related to the analysis of vertical profile data in Section 3.2. Finally, we adapt our refined approach to perform data integration based on the mixed modeling approach and relevant statistical principles, with the goal of producing a regional-scale vertical ozone profile product (Section 3.3).

### 3.1. Mixed Models for Multiple Correlated Data Sources

Mixed models in statistics are designed to combine different sources of data into two components (McLean et al., 1991): the first part represents the consensus underlying process (originally called the fixed effect), and the second part is the potential discrepancy resulting from the individual measuring procedures (originally called the random effect). This modeling approach assumes that even though some structured or unstructured discrepancy is expected from different measurement procedures, it seeks to determine a consensus process derived from the various sources of records.

Although the literature often uses the term "random effect" to distinguish it from the consensus process (Fisher, 1919), the specific cause of the discrepancy might not be actually random. For example, sparse sampling frequency is one of the primary reasons for data under-representativeness (Weatherhead et al., 2017). Especially for in situ observations in atmospheric sciences, we typically have too few data sources and limited samples to achieve a simple average that can reconcile all of the discrepancies. Therefore, the class of mixed models is the preferred approach, because the potential discrepancy from different monitoring stations can be treated separately.

In summary, the general relationship of the mixed models can be expressed as:

$$\text{data sources} = \text{consensus process} + \text{discrepancy} + \text{random noise}. \tag{2}$$

however, the type of data formulation also introduces additional complexity. For example, (a) when the linear correlation between the consensus process and the other data sources needs to be determined, linear mixed models use a fixed intercept and slope to represent the overall linear relationship, and also assign individualized intercept and/or slope to each data source for a better representation of data heterogeneity (Finney et al., 2016); (b) when regional variations need to be derived from multiple correlated time series measured at different locations, a single coefficient for intercept and/or slope might not be flexible or adequate, thus the regression spline can be

used to constitute the overall variations and also provide an adjustment to each individual time series (Chang et al., 2017; Wood et al., 2017). Whereas previous attempts were made for the incorporation of time series data on a horizontal surface, such as a regional or national surface ozone monitoring network (Chang et al., 2017), additional effort might be required for vertical profile data. We introduce the modeling of such vertical correlated structures in the next section.

### 3.2. Trend and Anomaly Detections for Vertical Profile Data From an Individual Data Source

The statistical framework of trend detection for ozone vertical profile data was described by Chang et al. (2020). An important consideration of trend detection for vertically distributed time series is that the trends should not be isolated to a single narrow pressure level or layer (e.g., a depth of 10 hPa); rather we expect to observe similar trends in the neighboring pressure layers as well. Therefore, the idea is to investigate similar structures in the time series from neighboring pressure levels to identify signal and noise components of the data, that is, borrowing the correlated variation to achieve a better representation of ozone systematic variation. In simple terms, this procedure can be thought of as boosting the sample size on a given pressure level. This idea can be implemented through statistical regularization, which is an efficient tool to filter out unstructured variations in the data (Poggio et al., 1987). However, to facilitate inter-comparison and combination of different data sources, and to enable anomaly detection (a different goal from trend detection, but it shares the same consideration discussed above), further adjustments to this framework need to be made for accommodating a broader range of data heterogeneity.

We summarize three major adjustments as follows:

1. Aligning data to common pressure levels: Ozonesonde and lidar instruments operated at different laboratories or institutions provide data at different vertical coordinates and resolutions. In order to make each data set comparable, we need to align the data to common vertical coordinates. We thus aggregate data into 10 hPa resolution layers, for example, the value at 500 hPa is the average of all available data over (495, 505) hPa. To avoid the influence of the decreasing pump efficiency and freezing/evaporation of sensing solutions near the top of the profile, we set the upper limit of this analysis to 10 hPa (including any data within (10,15) hPa but not above 10 hPa). Although the boundary is extended to 10 hPa, we will only place our focus on tropospheric ozone trends and anomalies
2. Deseasonalizing and normalizing data series at each pressure level: We assume that meaningful anomalies should not be limited to a single narrow pressure level, so carrying out the anomaly detection for multiple neighboring pressure layers simultaneously is desirable for vertical profile data. However, the magnitude of ozone variability (both mean and variance) in the upper troposphere/lower stratosphere can be much stronger than the middle or lower troposphere, and this heterogeneity introduces another layer of complexity to the anomaly detection problem

To consistently detect ozone anomalies across the entire profile, we need to make the variability across different levels more homogeneous. For each individual station, we first deseasonalized the data series by simply removing the long-term monthly climatology at a given pressure level. Even though seasonality is an essential component in the modeling of ozone time series, different approaches to estimate seasonal cycle generally have a negligible impact on the estimation of other components in the statistical model (e.g., trend estimate and its uncertainty; Weatherhead et al., 1998). We then normalized the data by using the standard deviation (SD) of deseasonalized series at each pressure layer (the deseasonalized mean is expected to be zero). Deseasonalization allows us to focus on the deviations independent of the climatology, while normalization makes the data homogeneous and comparable (otherwise relatively low tropospheric values and relatively high stratospheric values act like outliers to each other). Therefore, the statistical models are applied to the normalized deviations (ND), however, the seasonal mean and SD on any pressure level can always be transformed back (to the units of ppbv) from those ND.

1. Statistical regularization for trend and anomaly detections: The methodology developed in Chang et al. (2020) can be seen as a seasonal-trend decomposition (Cleveland et al., 1990) designed for multiple (vertically) correlated time series. The penalized regression splines (Wood, 2006) are applied to decompose the vertical profile time series data according to their seasonal patterns and interannual changes. Instead of deriving interannual changes based on annual mean normalized values at each pressure layer, in this study we directly perform the anomaly detection on monthly (aggregated) ND from each data set, in order to investigate and enhance the delicate variation structures at sub-seasonal scale. Indeed, Tarasick et al. (2019) concluded that

there was a modest bias (5%–8%) between IAGOS data and ozonesondes (i.e., mean values of ozonesonde data are higher). This bias is likely to endanger trend analysis, but by using the ND for each data source, this bias is expected to be removed before we combine different data sets in the next section

In terms of anomaly detection for the ND, the empirical rule of a normal distribution states 68%, 95% and 99% of data falls within $\pm 1$, $\pm 2$ and $\pm 3$ ND, respectively, so the strong anomalies of a single (normally distributed) time series can be identified as the magnitude of normalized values beyond $|\pm 2|$ or $|\pm 3|$ (Aggarwal, 2015). However, since we use the regularization technique, we expect that (a) any single point spike tends to be filtered out by the regularization, but an episode of a series of anomalies will be revealed in the model fitted result. This feature is thus desirable for our goal of anomaly detection from multiple time series; and (b) the fitted deviations should not exhibit such extreme values, for example, beyond $|\pm 2|$ or $|\pm 3|$ (since this anomaly detection is based on the fused version of multiple means), unless there was a highly exceptional (tipping) event that caused an episode of a long series of highly extreme anomalies. Given that the COVID-19 economic downturn progressed over many weeks in different regions of the world (Gkatzelis et al., 2021; Le Quéré et al., 2020a, 2020b), and given that tropospheric ozone has a lifetime of several weeks (Monks et al., 2015), we do not expect the ozone changes due to the economic downturn to be abrupt and dramatic (i.e., with a strong magnitude of anomaly), but rather to have transitioned gradually.

### 3.3. Data Incorporation for Combining Vertical Profile Data

In the previous section we discussed the methodology of anomaly and trend detections of vertical profile data at an individual data source, now the focus is placed on how to combine vertical profile data from multiple data sources. The regression splines applied in Chang et al. (2020) are fitted through the framework of generalized additive models (GAM; Hastie & Tibshirani, 1990). Combined with the mixed models described in Section 3.1, this extended framework of models is called (Wood, 2006). Let $y_s (h, t)$ represent the ozone measured at pressure level $h$, with a temporal index $t$, from station $s$, then the statistical model can be expressed as:

$$y_s(h, t) = f(h, t) + g_s(h, t) + \epsilon, \tag{3}$$

where $f (h, t)$ is the consensus process which represents the underlying ozone vertical distribution evolving with time (only related to $h$ and $t$); $g_s (h, t)$ represents the structured discrepancy (if any) from station $s$ in addition to the consensus process; and $\epsilon$ is the random noise.

At this point, the above consideration treats each station (or each source of data records) as equally important. In order to take the sampling frequency and inherent uncertainty in each data source into account, we derive the uncertainty estimate associated with each monthly normalized mean deviation using the following procedures:

1. We calculate the monthly standard error (the monthly SD divided by the number of profiles in that month) at each pressure surface for each data source, so the larger the data variability and/or the smaller sample size, the larger the standard error
2. Since the ozone variability from one pressure surface to another can be high, especially when comparing tropospheric to stratospheric pressure surfaces, we divide the monthly standard error by the mean ozone value on the corresponding pressure surface (i.e., taking the average of all ozone values on a particular pressure surface from the full period, determined by each data set), so these uncertainty estimates are more homogeneous and can be compared across different layers (analogous to the purpose of normalization). We refer to this quantity as the "relative variability"
3. Another complexity is that, similar to the monthly normalized mean deviations, the relative variability can be very noisy, and directly utilizing these quantities as model weights can result in noisy and unstable output. Therefore, we apply the penalized regression splines to the relative variability, in order to obtain a more consistent representation of uncertainty associated with each data record

We use the result from the above procedures to represent the uncertainty estimates, so every aggregated monthly mean deviation from each data source will have an associated uncertainty estimate. We use the inverse of the squared uncertainty as the weight in the model fitting process (Aitken, 1936), thus a data source with a higher sampling frequency and/or lower variability receives a higher weight.

The model fitting procedure involves the balance of minimization of (weighted) model residuals and multiple roughness penalties (also known as the regularization), which aims to avoid overfitting to unstructured variations. Overall, the proposed model can be solved by minimizing the penalized least square criterion (Wood, 2006):

$$\left\| \sqrt{W} \left( y_s(h,t) - f(h,t) - g_s(h,t) \right) \right\|^2 + D^2\mathbf{F}. \tag{4}$$

where $\| \cdot \|$ is the Euclidean norm, $W$ is the weight matrix, $D^2\mathbf{F}$ is the roughness penalty term (see supplementary material for details), and $f(h,t)$ and $g_s(h,t)$ are determined through basis representations:

$$f(h,t) = \sum_{i=1}^{N} p_i X_i(h,t) \text{ and } g_s(h,t) = \sum_{j=1}^{M} q_{s,j} Z_j(h,t), \tag{5}$$

where $p_i$ and $q_{s,j}$ are unknown coefficients associated with two dimensional basis functions $X_i(h,t)$ and $Z_j(h,t)$. The correlation structures at neighboring pressure surfaces and temporal correlations are determined by the Duchon splines (Wood et al., 2017). To produce high resolution sub-seasonal structure (e.g., for investigating when the 2020 anomalies become pronounced), for $f(h,t)$ the basis dimension is set to the maximal capacity (e.g., $N = 2000$), respectively, for below 750 hPa, 500–750 hPa, 250–500 hPa and above 250 hPa. The specification of $g_s(h,t)$ is currently not a crucial part of the model; this term aims to inform the model about the source of data, as the model does not recognize that a set of data is measured from the same station. The implication of the penalized least square in Equation 4 is that the model fitting is not solely dependent on the weighting scheme of each data source, but on the joint consideration of consensus variability between different data sets, sampling uncertainty within each data set, and a measure to avoid overfitting.

The mathematical background and algorithms involved in solving the GAM/GAMM are widely discussed in the literature (e.g., Hastie and Tibshirani (1990); McLean et al. (1991); Pinheiro and Bates (2006); Ruppert et al. (2003); Wood (2006)). The framework of trend and anomaly detections for vertical profile data is implemented through the *R* package mgcv (Wood, 2006), and the implementation details are described by Chang et al. (2020).

## 4. Results

The GAMM methodology described above is now used to calculate the regional-scale ozone vertical distributions, anomalies and trends above Western Europe (Section 4.1) and western North America (Section 4.2), with a summary of the 2020 regional anomalies above these two regions presented in Section 4.3.

### 4.1. Regional Ozone Above Western Europe

#### 4.1.1. Derive the Fused Ozone Product

A total of nine vertical ozone profile records are available above Western Europe with data extending back to 1994 (Table 1). Sampling above the site of Lindenberg, Germany has been rather limited with a large data gap from 2014 to 2019 (Figure S4 in Supporting Information S1) and this site is not considered further. We first demonstrate the ozone variability above each individual site, with Figure 2 showing the monthly aggregated time series data in the free troposphere (700-300 hPa) and the corresponding deseasonalized series (by simply subtracting the long-term monthly means at a given pressure level) and associated standard errors. Each station samples a different part of Europe and has a different sampling frequency with different days and times. These factors introduce different data variability. Even though these time series may look similar (panel (a)), they become less comparable when the regular part (i.e., seasonal cycle) of the data has been removed (panel (b)), and we see that the uncertainties behind the monthly aggregations are very different (panel (c)). The IAGOS data have the lowest standard error values and therefore the lowest uncertainty, followed by the ozonesonde records at Uccle, Payerne and HPB, with the greatest uncertainty observed above De Bilt, OHP, Legionowo and Madrid; this result is expected and matches the sampling frequency of each data set (see Section 2.1 and Figure 1).

The following analysis places the focus on the modeling of deseasonalized and normalized ozone deviations. In this analysis we aim to fuse IAGOS data with ozonesonde data measured from Uccle, Payerne, De Bilt, OHP and HPB, as these sites are in close spatial proximity above western Europe (Legionowo and Madrid are more than

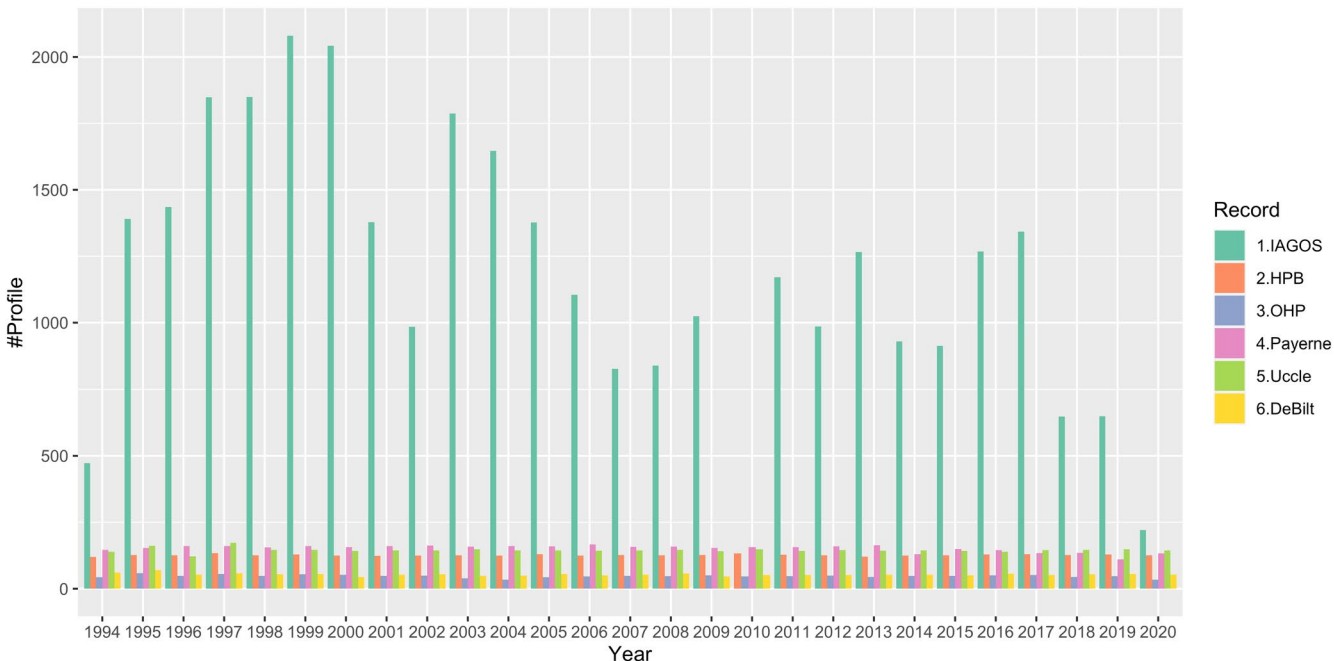

**Figure 1.** Comparisons of annual sample sizes from In-Service Aircraft for a Global Observing System and ozonesonde data sets in Europe.

800 km from the five sites that are in close proximity). The total area covered by these six data sets is 7° latitude by 7° longitude, or 400,000 km².

Here we focus on the ozone distribution derived from the IAGOS data set, and the comparison of the derived consensus process of five ozonesonde records without and with the IAGOS data set (hereinafter referred to as the intermediate and the final "fused" data products, respectively). The results are shown in Figure 3 for the distributions based on the ND, and in Figure 4 for the distributions transformed back to units of ppbv.

The final fused product provides our best estimate of ozone variability above a region of western Europe that covers approximately 400,000 km², and therefore some of the interannual variability will be impacted by variations in sampling frequency across the region. The fused product is based on more than 45,000 ozone profiles during the 27-year period from 1994 to 2020, which corresponds to an average sampling frequency of approximately 140 profiles per month. Our previous work showed that a sampling frequency of at least 18 profiles per month is required for accurate trend quantification above a single site (Chang et al., 2020), when conducting a simple analysis of data on isolated pressure surfaces. However, a sampling frequency of 14 profiles per month is adequate when applying our statistical regularization method, which takes advantage of correlated structures on neighboring pressure surfaces. While our fused product is impacted by spatial variability, which is not an issue for individual sites, it contains 10 times the data necessary for accurate trend quantification at a single site, and this product provides an extremely detailed view of ozone interannual variability and long-term trends above Western Europe. Overall, the final fused product reveals many details regarding the interannual variability of ozone above Europe, which cannot be detected by any individual ozonesonde data set due to the relatively low sampling frequency. Even the IAGOS data set, which comprises 70% of the total ozone profiles, does not provide a full representation of ozone's interannual variability due to the large data gap in 2010 and diminished sampling in 2020. However, our data fusion approach takes advantage of the strengths of each data set to provide the most complete ozone record possible, given current data limitations.

A noticeable change from the intermediate product to the final fused product (with IAGOS data included) can be observed in the mid-1990s, which may cause concern because the trend estimates could be sensitive to the first and last few years of the study period; this discrepancy between IAGOS and ozonesonde data in the years before 1997 was previously reported by Logan et al. (2012), and will be discussed in the next section. Another notable feature is the strong positive anomaly in the free troposphere that occurred in 1998–1999, previously attributed to

**Table 2**
*Ozone Trends in the Free Troposphere (700–300 hPa) Since 1994 (Some Stations Have Different Reference Years, and the TMF Lidar Trends are Limited to 600−−300 hPa)*

| Site | Reference | Through 2019 | | Through 2020 | | Change (%) |
|---|---|---|---|---|---|---|
| | | Trend (±2-$\sigma$) | *p*-value | Trend (±2-$\sigma$) | *p*-value | |
| Edmonton | 1994 | *−3.87 (±2.29)* | <0.01 | *−3.79 (±2.13)* | <0.01 | 2 |
| Kelowna/Port Hardy | 2003 | 1.54 (±2.31) | 0.18 | 1.21 (±2.07) | 0.25 | −22 |
| Trinidad Head (THD) | 1997 | −1.38 (±1.91) | 0.15 | **−1.89 (±1.80)** | 0.04 | −37 |
| Boulder | 1994 | *−1.38 (±0.79)* | <0.01 | *−1.37 (±0.75)* | <0.01 | 1 |
| Table Mountain (TMF) | 2002 | 1.07 (±2.21) | 0.33 | 0.58 (±2.00) | 0.56 | −45 |
| IAGOS (WNA) | 1994 | *3.97 (±1.80)* | <0.01 | – | – | – |
| Fused (WNA) | 1994 | *0.35 (±0.21)* | <0.01 | 0.14 (±0.21) | 0.19 | −61 |
| Legionowo | 1994 | *−1.52 (±1.02)* | <0.01 | *−1.60 (±0.98)* | <0.01 | −5 |
| Lindenberg | 1994 | −1.73 (±2.01) | 0.09 | *−2.85 (±1.83)* | <0.01 | −65 |
| De Bilt | 1994 | *2.26 (±1.04)* | <0.01 | *1.86 (±1.02)* | <0.01 | −18 |
| Uccle | 1994 | *1.49 (±0.89)* | <0.01 | **1.00 (±0.90)** | 0.03 | −33 |
| Hohenpeissenberg (HPB) | 1994 | −0.17 (±0.73) | 0.63 | −0.48 (±0.75) | 0.20 | −173 |
| Payerne | 1994 | *−1.56 (±0.85)* | <0.01 | *−1.81 (±0.83)* | <0.01 | −16 |
| Haute Provence (OHP) | 1994 | **1.29 (±1.13)** | 0.02 | 1.03 (±1.07) | 0.06 | −20 |
| Madrid | 1994 | −0.39 (±1.34) | 0.56 | −0.88 (±1.29) | 0.18 | −124 |
| IAGOS (Europe) | 1994 | *1.16 (±0.77)* | <0.01 | *1.02 (±0.75)* | 0.01 | −12 |
| Fused (Europe) | 1994 | *0.65 (±0.19)* | <0.01 | *0.36 (±0.20)* | <0.01 | −44 |
| Broadmeadows | 1999 | −1.07 (±1.26) | 0.09 | −0.71 (±1.18) | 0.23 | 34 |
| Lauder | 1994 | 0.34 (±0.77) | 0.38 | 0.27 (±0.73) | 0.46 | −21 |
| Macquarie Island | 1994 | *−4.23 (±1.96)* | <0.01 | *−3.74 (±1.85)* | <0.01 | 11 |
| Tateno | 2009 | −0.58 (±6.32) | 0.85 | −2.15 (±5.36) | 0.42 | −269 |
| Izaña | 1995 | *3.01 (±1.00)* | <0.01 | *2.65 (±0.97)* | <0.01 | −12 |
| Hong Kong | 2000 | 0.54 (±2.66) | 0.69 | 0.55 (±2.45) | 0.66 | 1 |
| Hilo | 1994 | 0.75 (±1.45) | 0.30 | 0.67 (±1.35) | 0.32 | −11 |

*Note.* Trend values and 2-sigma uncertainty (in units of ppbv/decade) are based on monthly means and linear regression models. The relative change (%) of trends is based on the absolute value of trend value through the end of 2019. Trends with a magnitude greater than 2-sigma are shown in bold font, and trends with a magnitude greater than 3-sigma are shown in bold and italic. The ozonesonde records at kelowna and port hardy are combined (see discussion in Section 2.1). The fused trends represent results derived from a regional data fusion of available data records above western North America (WNA) or Europe (see Section 4)

enhanced stratosphere-troposphere exchange, biomass burning and transport from Asia following the very strong El Niño event of 1997/1998 (Koumoutsaris et al., 2008).

Further details of this analysis are provided in the supplementary material, including:

1. Figure S3 in Supporting Information S1 shows the diagnostics of statistical model fitting of this final fused product. The residuals from the fitted model are plotted as a histogram and as a function of each month, year and monitoring station. Since the distributions of residuals are centered around zero in each scenario, the model provides a good representation of the mean distribution
2. Figures S4 and S5 in Supporting Information S1 show the vertical ozone distribution above each station, including the other three European stations that were not included in this analysis, that is, Madrid, Lindenberg and Legionowo. Any blank in the curtain plots indicate that measurements are unavailable (e.g., IAGOS data above 200 hPa). Due to the fact that the regular part of the data has been removed and each data record was sampled from different locations and times, these curtain plots show the remaining interannual and vertical variability that can be captured by the methodology

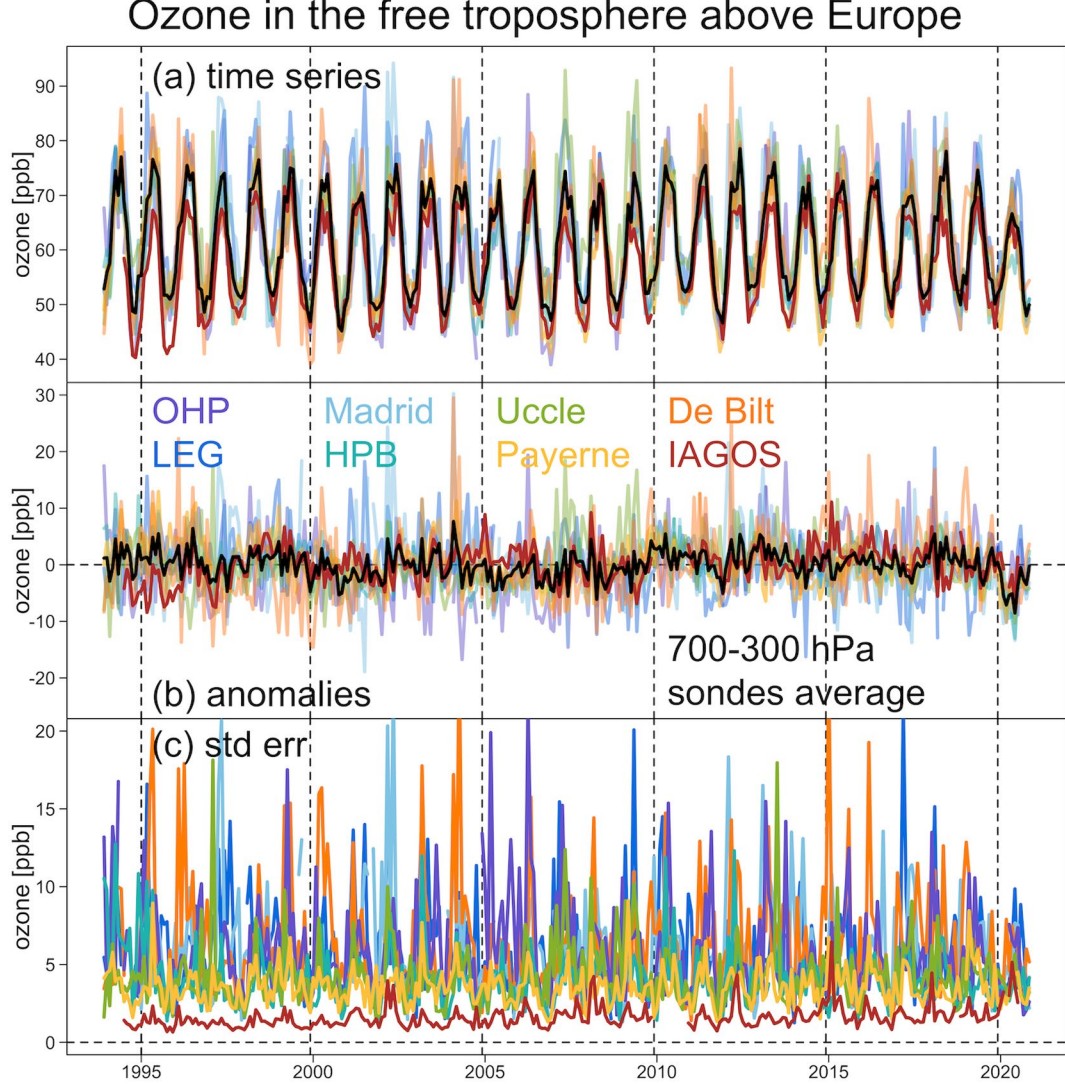

**Figure 2.** Free tropospheric ozone (a) observations, (b) deseasonalized anomaly series, and (c) standard error series (in units of ppbv) above Europe (1994–2020). Black curves represent the simple averages of ozonesonde records (and not including In-Service Aircraft for a Global Observing System data set).

3. Figure S6 in Supporting Information S1 shows the relative variability that we used to weight the data, with the high frequency IAGOS data receiving the greatest weighting, and the low frequency Haute-Provence and De Bilt data receiving the lowest weighting

4. Note that the curtain plot from each individual station is independent from the fused product. The regression splines are applied twice in the data fusion process: the first time is to obtain the uncertainty estimate in each data set (Figure S6 in Supporting Information S1), and the second time is to fuse multiple data sets according to their data uncertainty (Figures 3 and 4). It should also be noted that we tried applying the same high resolution setting to the IAGOS and each individual ozonesonde data set, as applied to the final fused product, but the results could not produce fine scale structures similar to the final fused product (unless a much longer data record is available, as demonstrated for the 50-year Uccle record in Figure S9 in Supporting Information S1). Therefore, the details of the final fused product highlight the benefit of incorporating all available data sources in a relatively shorter time frame

5. Previous studies found that stratospheric air masses play an important role in deriving the ozone trends in the upper troposphere (Chang et al., 2020; Gaudel et al., 2020). Figure S7 in Supporting Information S1

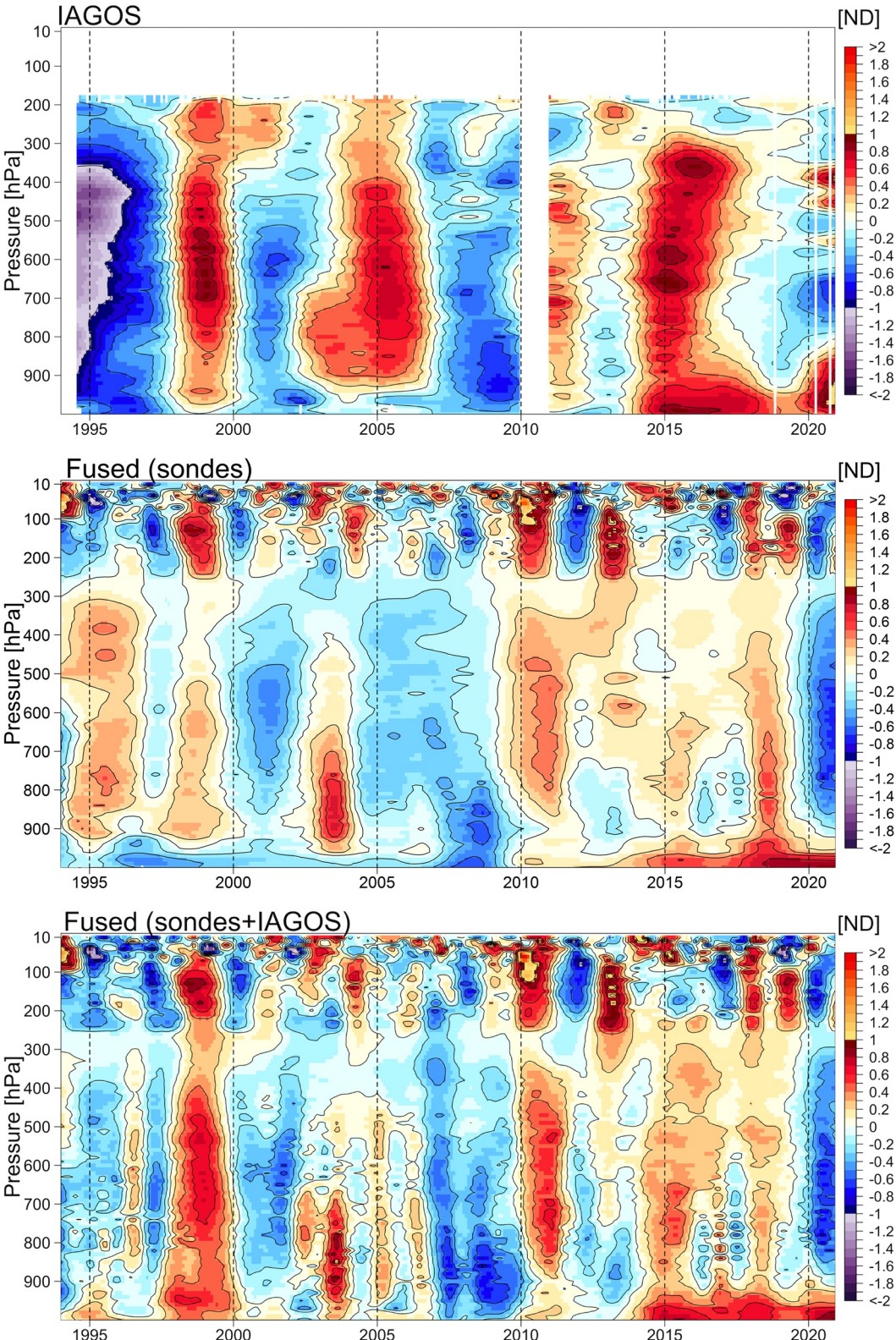

**Figure 3.** Ozone mean distributions above Europe based on normalized deviation. Panels show the results of In-Service Aircraft for a Global Observing System data set (top), intermediate fused product (middle), and final fused product (bottom).

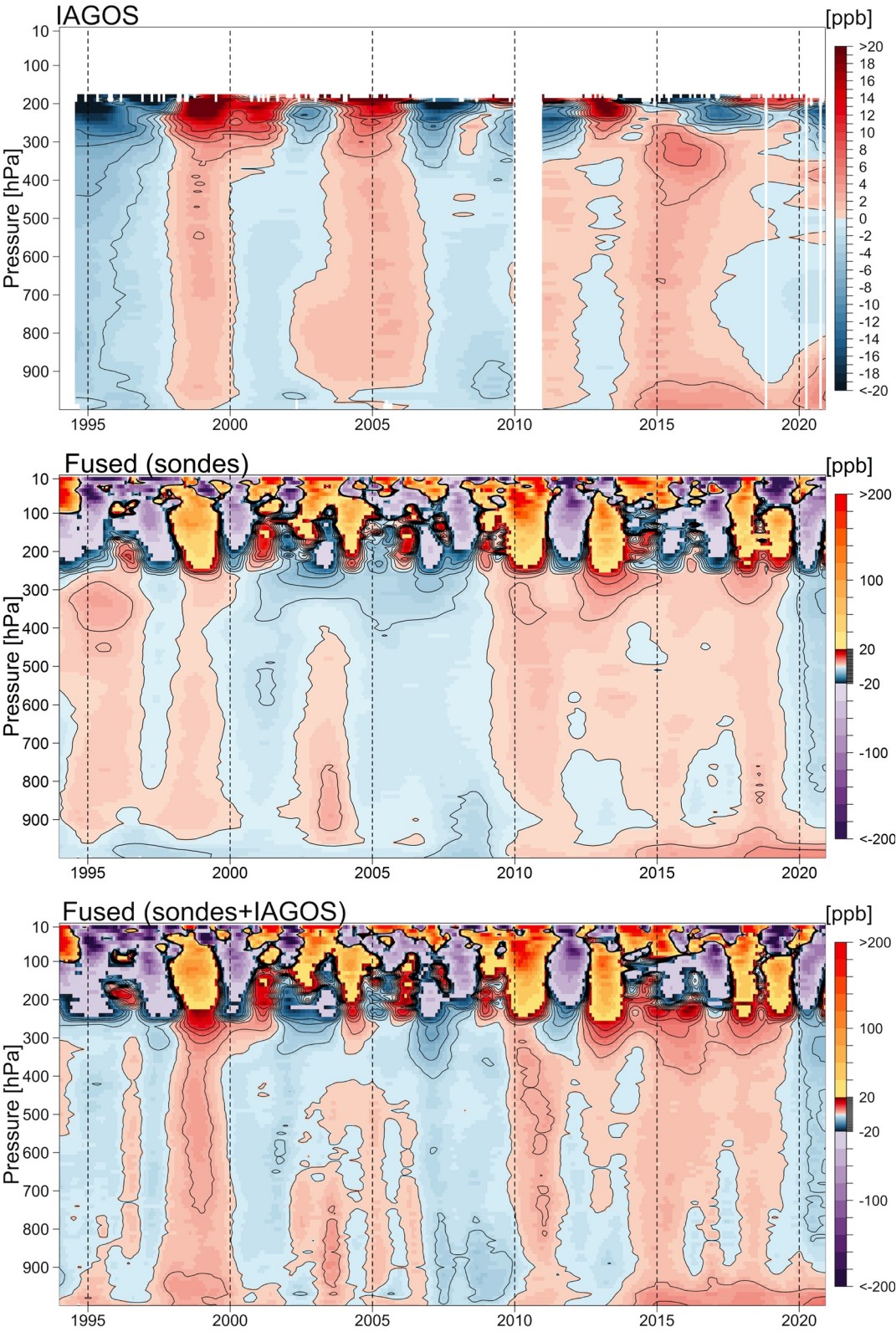

**Figure 4.** Same as Figure 3, but ozone mean distributions are transformed back to the units of ppbv.

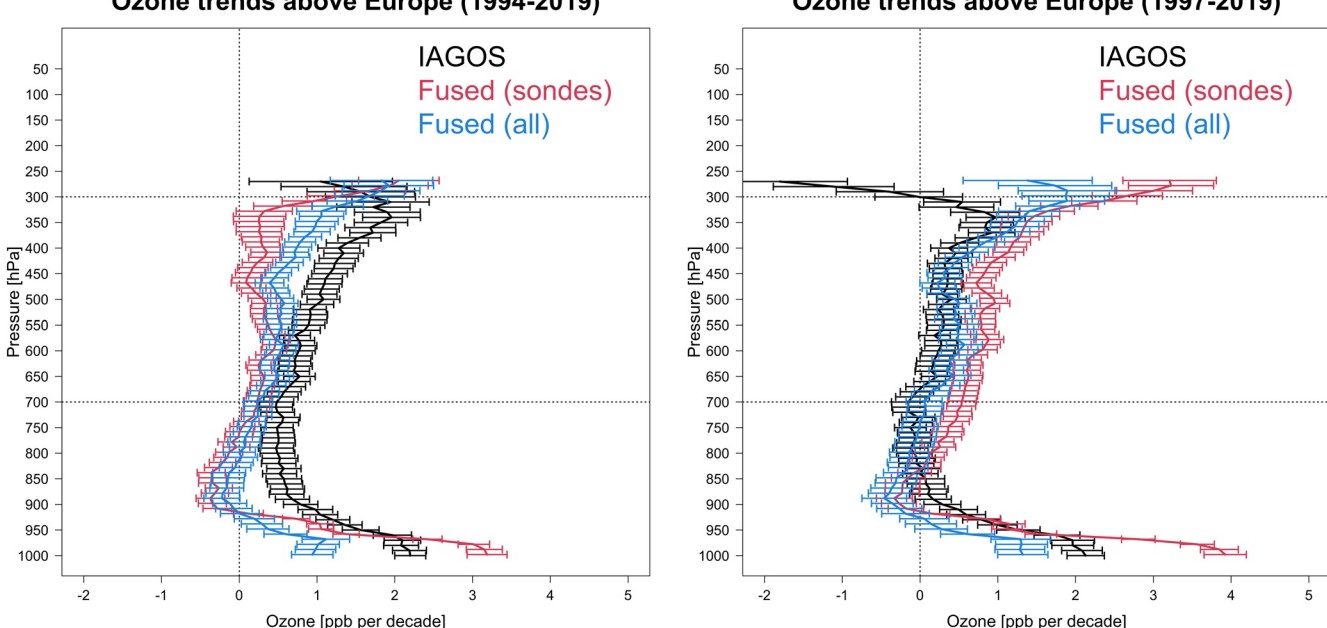

**Figure 5.** Profiles of ozone mean trends above Europe (in units of ppbv/decade) derived from In-Service Aircraft for a Global Observing System, intermediate and final fused products over 1994–2019 (top left panel) and 1997–2019 (top right panel).

demonstrates that when the stratospheric air masses are filtered (which are determined from the potential vorticity values provided by the IAGOS data portal), the magnitude of trends in the upper free troposphere (300 hPa) are reduced by about 50%, but the impact on middle and lower free tropospheric ozone trends is weak. However, in order to maintain the compatibility between IAGOS and ozonesonde data sets, stratospheric air masses are not filtered from the IAGOS data in this study

6. Even though the trend estimates are diverse at individual sites, the highly sampled ozonesonde data can provide unparalleled insight into how ozone has changed over very long time periods (e.g., 50 years). Figure S9 in Supporting Information S1 shows the ozone vertical distributions above Uccle over 1969–2020 (Van Malderen et al., 2021), which clearly reveals the year-to-year tropospheric ozone variability over 50 years. As discussed previously, fine scale vertical structures are generally difficult to identify at individual stations (see Figure S4 in Supporting Information S1), because of a lack of sufficient samples to pin down the complex variation. In contrast, with several decades of data and a high sampling frequency, the fine scale structure can be produced at a single location. The Uccle record serves as an important reference for the historical context of tropospheric ozone changes, since very few monitoring stations have maintained continuous operation of such high frequency long-term measurements over a half century

### 4.1.2. Evaluate the Fused Ozone Product

In terms of trend detection, to account for the potential impact of ENSO and QBO (features correlated with these dynamical phenomena can be seen in the fused record above 250 hPa in Figure 4), we use the regression model in Equation 1 to derive the trend estimates (Neu et al., 2014; Weatherhead et al., 2000; Ziemke et al., 1997). In this study we do not report the trends above 250 hPa.

We derive the trend estimates on each 10 hPa surface from the curtain plots of IAGOS data, intermediate and final fused products referenced to January 1994 and 1997, respectively (Figure 5 and Table 3). A noticeable discrepancy between the IAGOS data set and the intermediate ozonesonde product is visible in the aggregated time series between August 1994 and December 1996 in Figures 2–4 (this discrepancy was also reported by Logan et al. (2012) and Staufer et al. (2014)). The IAGOS anomalies tend to be strongly negative, while the anomalies from the intermediate product tend to be positive (with respect to the period 1994–2019); this discrepancy thus produces different trend results. When data from 1994 to 1996 are omitted, we find that the trends in the ozone-sonde product become stronger, and the trends in the IAGOS product become weaker, due to the sensitivity of the trends against different reference years. However, the final fused product (IAGOS and ozonesondes) changes

**Table 3**
*Comparisons of Trends (in Units of Ppbv/Decade) Based on the Integrated Fit (With the Vertical Correlations Accounted for) and Derived From the IAGOS, Intermediate and Final Fused Products Above Europe Referenced to the Year 1994 or 1997*

| Site | Pressure (hPa) | 1994–2019 | | 1994–2020 | | Change (%) |
|---|---|---|---|---|---|---|
| | | Trend (±2σ) | *p*-value | Trend (±2-σ) | *p*-value | |
| IAGOS | 950–250 | *0.97 (±0.25)* | <0.01 | *0.88 (±0.23)* | <0.01 | −10 |
| | 700–300 | *1.08 (±0.24)* | <0.01 | *0.96 (±0.22)* | <0.01 | −11 |
| | 400–300 | *1.67 (±0.33)* | <0.01 | *1.54 (±0.31)* | <0.01 | −8 |
| | 650 | *0.72 (±0.21)* | <0.01 | *0.54 (±0.21)* | <0.01 | −25 |
| | 950–800 | *0.89 (±0.21)* | <0.01 | *0.94 (±0.20)* | <0.01 | 7 |
| Fused (intermediate) | 950–250 | *0.35 (±0.19)* | <0.01 | 0.11 (±0.19) | 0.27 | −70 |
| | 700–300 | *0.34 (±0.20)* | <0.01 | 0.06 (±0.21) | 0.59 | −83 |
| | 400–300 | *0.46 (±0.32)* | <0.01 | 0.09 (±0.32) | 0.57 | −80 |
| | 650 | *0.29 (±0.16)* | <0.01 | 0.06 (±0.17) | 0.48 | −80 |
| | 950–800 | −0.03 (±0.17) | 0.74 | −0.09 (±0.16) | 0.23 | −235 |
| Fused (final) | 950–250 | *0.60 (±0.20)* | <0.01 | *0.35 (±0.20)* | <0.01 | −42 |
| | 700–300 | *0.65 (±0.19)* | <0.01 | *0.36 (±0.20)* | <0.01 | −44 |
| | 400–300 | *1.08 (±0.27)* | <0.01 | *0.66 (±0.29)* | <0.01 | −39 |
| | 650 | *0.47 (±0.18)* | <0.01 | *0.25 (±0.19)* | 0.01 | −47 |
| | 950–800 | −0.03 (±0.21) | 0.81 | −0.05 (±0.20) | 0.61 | −99 |

| Site | Pressure (hPa) | 1997–2019 | | 1997–2020 | | Change (%) |
|---|---|---|---|---|---|---|
| | | Trend (±2-σ) | p-value | Trend (±2-σ) | p-value | |
| IAGOS | 950–250 | 0.19 (±0.23) | 0.09 | 0.14 (±0.21) | 0.20 | −30 |
| | 700–300 | *0.34 (±0.22)* | <0.01 | *0.26 (±0.21)* | 0.01 | −26 |
| | 400–300 | *0.70 (±0.32)* | <0.01 | *0.62 (±0.30)* | <0.01 | −12 |
| | 650 | 0.15 (±0.15) | 0.15 | −0.02 (±0.21) | 0.85 | −113 |
| | 950–800 | *0.42 (±0.22)* | <0.01 | *0.52 (±0.21)* | <0.01 | 25 |
| Fused (intermediate) | 950–250 | *0.89 (±0.20)* | <0.01 | *0.53 (±0.22)* | <0.01 | −41 |
| | 700–300 | *1.00 (±0.20)* | <0.01 | *0.58 (±0.23)* | <0.01 | −42 |
| | 400–300 | *1.78 (±0.42)* | <0.01 | *1.00 (±0.21)* | <0.01 | −44 |
| | 650 | *0.61 (±0.18)* | <0.01 | *0.29 (±0.20)* | 0.01 | −53 |
| | 950–800 | 0.12 (±0.21) | 0.25 | 0.03 (±0.19) | 0.79 | −80 |
| Fused (final) | 950–250 | *0.51 (±0.25)* | <0.01 | 0.20 (±0.25) | 0.12 | −62 |
| | 700–300 | *0.63 (±0.24)* | <0.01 | *0.26 (±0.25)* | 0.04 | −59 |
| | 400–300 | *1.44 (±0.41)* | <0.01 | *0.63 (±0.24)* | <0.01 | −56 |
| | 650 | *0.37 (±0.22)* | <0.01 | 0.09 (±0.23) | 0.42 | −75 |
| | 950–800 | −0.19 (±0.27) | 0.16 | −0.20 (±0.24) | 0.10 | −8 |

little when the 1994–1996 data are omitted (Figure 5 and Table 3). Overall, mid-tropospheric ozone above Western Europe increased at the rate of $0.65 \pm 0.19$ ppbv decade$^{-1}$, from 1994 to 2019, equal to a total increase of $1.6 \pm 0.5$ ppbv, or ~3%.

Note that the results presented in Table 3 are based on our methodology that accounts for the vertical correlations, referred to as the integrated method in Chang et al. (2020), and the results for the individual data sets in Table 2 are based on the simple separated fit that does not account for the vertical correlations.

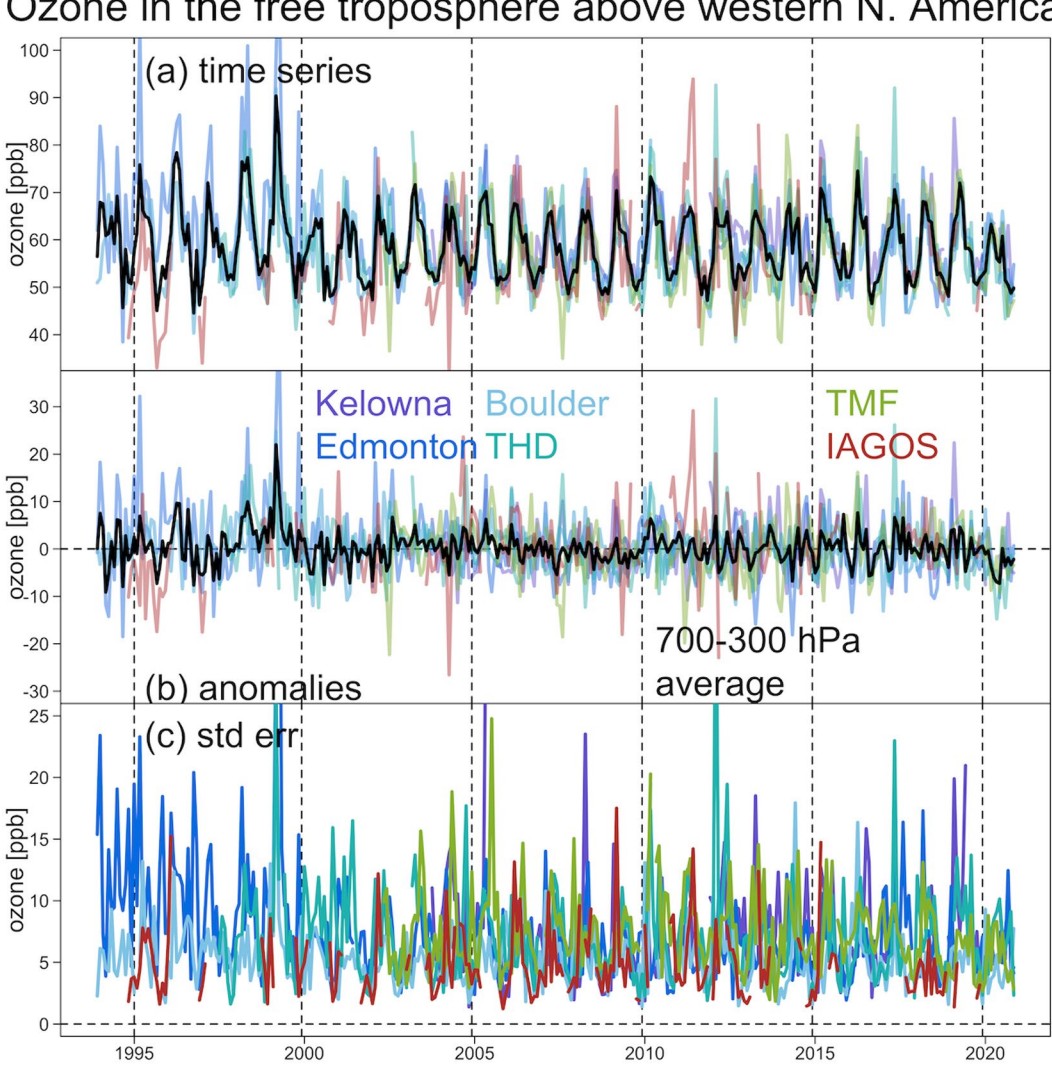

**Figure 6.** Free tropospheric ozone (a) observations, (b) deseasonalized anomaly series, and (c) standard error series (in units of ppbv) above western North America (1994–2020). Black curves represent the simple averages of ozonesonde, lidar and In-Service Aircraft for a Global Observing System records.

## 4.2. Regional Ozone Above Western North America

The study region of western North America is comprised of six measurement sites across an area spanning 19° latitude by 19° longitude (3,200,000 km²), which is 8 times larger than the area of the fused product above western Europe. This study region is similar to the region analyzed by Cooper et al. (2010), who focused on April-May 1995–2008 to show that ozone had increased during springtime above western North America, and that the rate of increase was stronger for air masses that had experienced direct transport from Asia. Cooper et al. (2010) conducted a range of sensitivity tests to determine the impact of the large spatial scale of this region on the overall trend. They found that a reduction in the spatial scale of the sampling areas resulted in a positive trend similar to that of the full area. A further sensitivity test found that the relatively sparse sampling strategy across the region increased the uncertainty of the trend estimate (Lin et al., 2015). However, our updated analysis for this region spans a much longer period (1994–2020), which is expected to be more robust due to an additional decade of available data.

Figure 6 presents the factual ozone variability in the free troposphere above western North America, as reported by the individual ozonesonde, lidar and commercial aircraft time series. Since the sampling frequency from the IAGOS program is much more limited and sparse in North America, compared to Europe, the data uncertainty

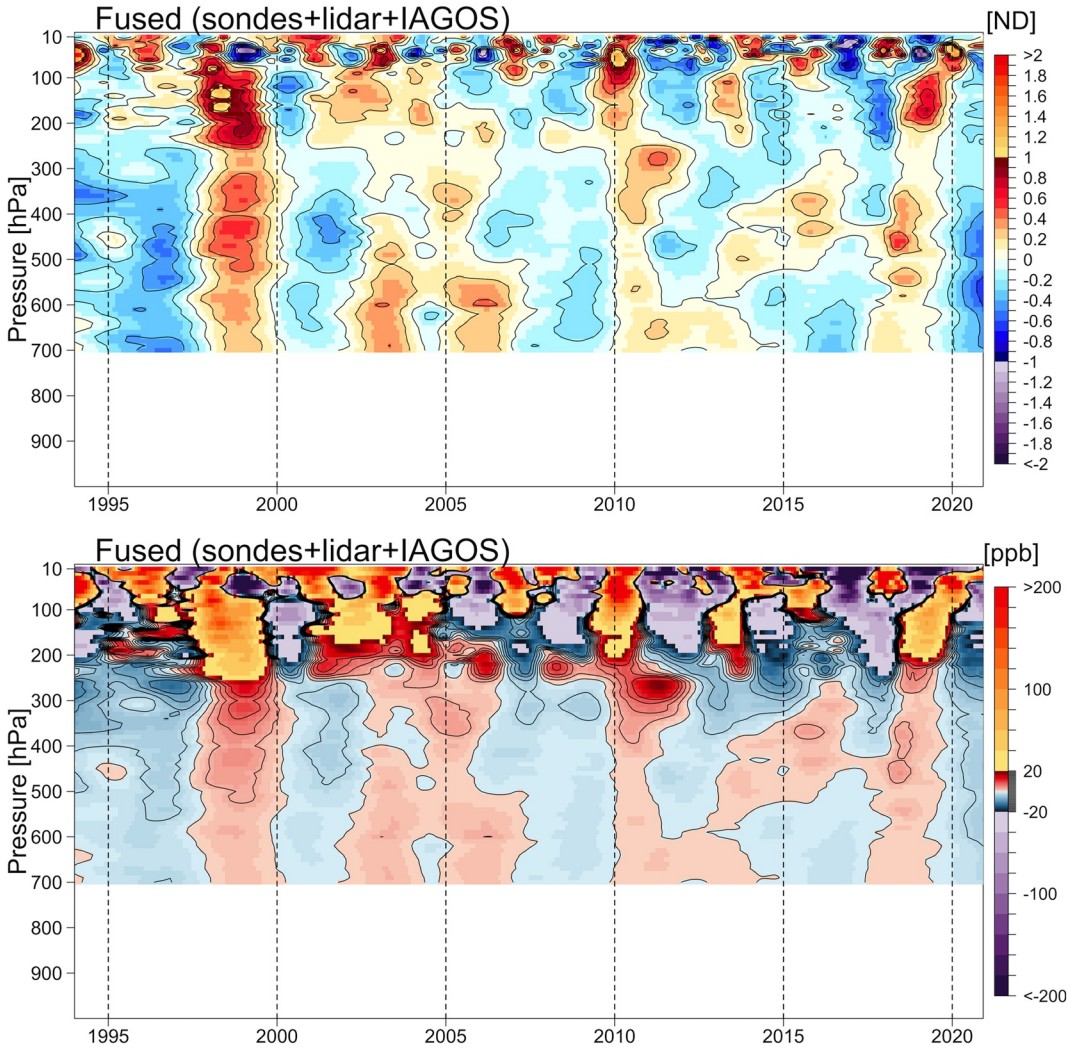

**Figure 7.** Fused ozone mean distributions of all available ozonesonde, lidar and In-Service Aircraft for a Global Observing System data above western North America based on normalized deviation and in units of ppbv.

is not as low as the European IAGOS data, thus the IAGOS data do not play a strong role in the data weighting process.

Since the monitoring stations in western North America are spread across a relatively large area with a range of urban, rural and marine environments in the boundary layer, we do not produce the fused ozone product below 700 hPa. Figure 7 shows the vertical distributions based on the ND and transformed back to the units of ppbv from the fused product of ozonesonde records from Boulder, THD, Edmonton and Kelowna/Port Hardy, lidar records from TMF and the IAGOS data set (no intermediate product is shown). The corresponding vertical distributions and uncertainty estimates for each individual station are provided in Figures S13-S15 in Supporting Information S1.

It is worth mentioning that even though the western North America fused product has the same number of in situ data sources as the European fused product, the overall sample size (∼9,900 profiles) is much lower than that in Europe (∼45,700 profiles; see Table 1). This difference in sample size might be the main reason why the fused product in Europe shows more profound and detailed interannual structures, while the variability above western North America is still rather indistinct between neighboring years.

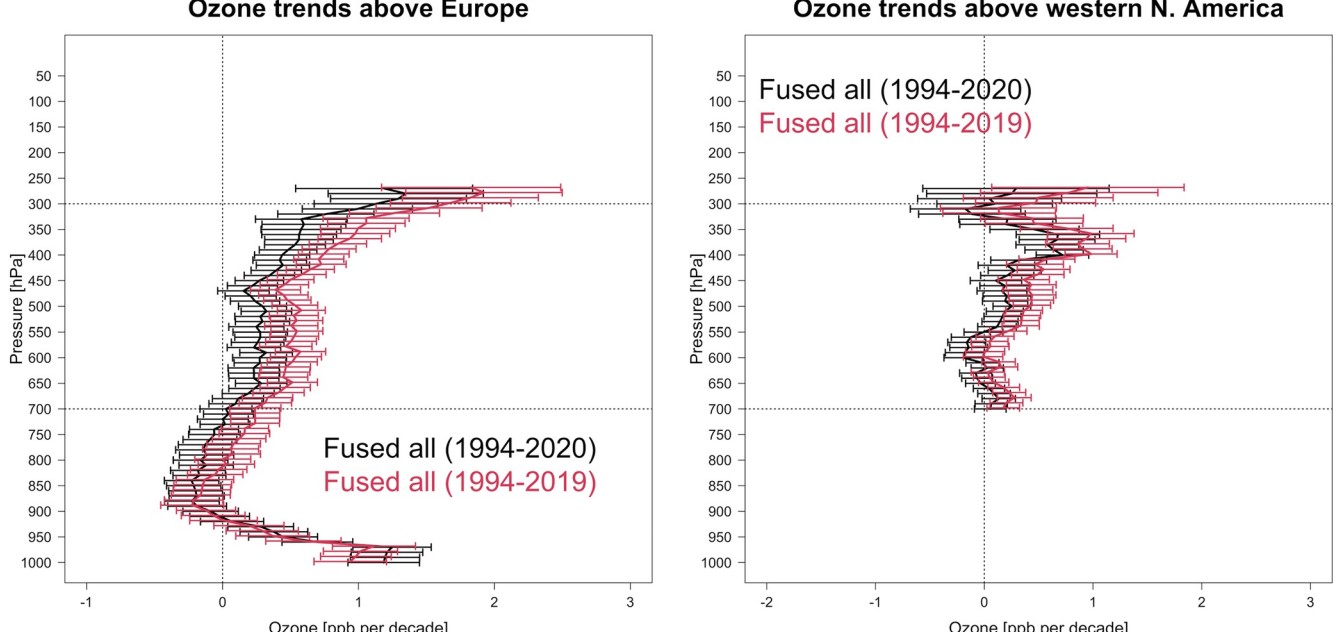

**Figure 8.** Profiles of ozone mean trends above Europe and western North America (in units of ppbv/decade) derived from the final fused product over 1994–2019 and 1994–2020.

### 4.3. Quantification of 2020 Anomalies and Impact on Trends Above Europe and Western North America

To summarize the 2020 regional ozone anomalies and their uncertainty above Europe and western North America, we first show the impact of 2020 anomalies on the long-term trends in Figure 8. In the lowermost pressure levels (below 950 hPa) in Europe, the trends are increasing incrementally in 2020 with respect to 1994–2019. This increase is consistent with a range of new studies that have shown surface ozone increased across many urban regions during the lockdown period (Gkatzelis et al., 2021; Sokhi et al., 2021). For both regions the distributions of trends from 850 hPa through 250 hPa are found to be consistently shifted toward lower or negative values in 2020. Overall, the estimated regional trends in the free troposphere above Europe decreased from 0.65 (±0.19, $p < 0.01$) ppbv/decade between 1994 and 2019 to 0.36 (±0.20, $p < 0.01$) ppbv/decade between 1994 and 2020, with a relative change of −44%. The corresponding trends in western North America decreased from 0.35 (±0.21, $p < 0.01$) ppbv/decade between 1994 and 2019 to 0.14 (±0.21, $p = 0.19$) ppbv/decade between 1994 and 2020, with a relative change of −61%.

Figure 9 compares the quantified 2020 anomalies in the free troposphere with respect to each individual year over 1994–2019. The 2-sigma uncertainty ranges are determined by the aggregated standard errors associated with the final fused product (as shown in Figures 4 and 7). The inverse-uncertainty weighted time series are also provided, in order to show that the unusual anomalies could be data points that substantially deviate from seasonal variations (i.e., those anomalies can be more noticeable after deseasonalization), and are not necessarily extreme values. Whereas the free-tropospheric analysis by Steinbrecht et al. (2021) only provided an average ozone anomaly value for spring and summer 2020, the uncertainty estimates provided in this study enable us to compare the detected regional anomalies with climatological values over 1994–2019 in an objective way. Based on our sophisticated statistical approach that detects regional anomalies and trends in the presence of data variability and uncertainty, Figure 9 provides robust insights: (a) Over the 27 years, only in 2020 do both Europe and western North America show substantial negative regional anomalies (the 2-sigma uncertainty range is apart from zero), and strong positive anomalies are only coincident above both regions in 1998, which has been attributed to enhanced STE following the strong 1997–1998 El Niño event (Cooper et al., 2010; Koumoutsaris et al., 2008; Langford, 1999; Thouret et al., 2006); (b) Even though localized variability is observed at individual stations, the overall variations from these two regions are deemed to be well correlated (Pearson correlation = 0.69); and (c) The overall 2020 quantified regional anomalies (with the overall trends over 1994–2019 removed/adjusted) in the

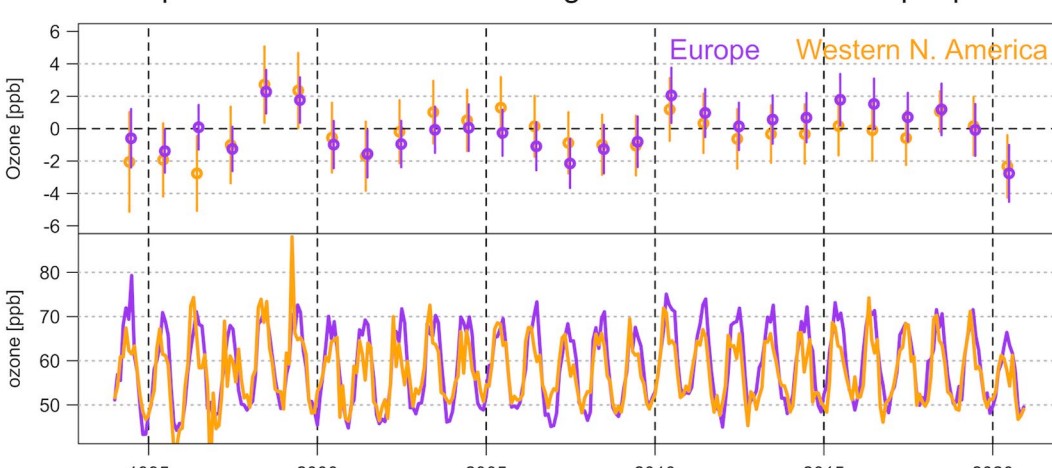

**Figure 9.** Quantified annual ozone mean anomalies (with 2-sigma intervals) and uncertainty weighted time series in the free troposphere (700-300 hPa) above Europe and western North America.

free troposphere are −3.60 (±1.75) ppbv above Europe and −2.77 (±1.92) ppbv above western North America, corresponding to percentage deviations of −6.0 (±2.9) % and −4.8 (±3.3) %, respectively.

The final analysis investigates the ozone variability within the year 2020. We fit the GAMM under the same setting previously, but limited to 2015–2020, to reveal additional fine scale structures in recent years (due to computational limitations, we cannot produce such detailed variability at the monthly scale over the full 27-year record). The resulting vertical distributions based on the ND are shown in Figure 10 for both regions. The economic downturn due to COVID-19 started between March and April, but the most profound anomalies can be observed around July in Europe, with a near-symmetric anomaly structure within the year 2020 in the free troposphere that does not extend above 300 hPa. Note that Figure 10 is based on ND at each pressure level, thus the absolute magnitude of quantified 2020 anomalies is deemed to be larger in the upper troposphere than the mid- and lower troposphere (Bouarar et al., 2021), when the vertical distribution is transformed back to the units of ppbv (see Figure S17 in Supporting Information S1). The strongest anomalies (700-300 hPa) above Europe occur in July with a magnitude of −4.60 ppbv (−7.5%), compared to a magnitude of −1.53 ppbv (−2.4%) in January and −1.88 ppbv (−3.1%) in December. A similar, but weaker anomaly structure can also be observed in the free troposphere above western North America. Averaged across northern mid-latitudes, tropospheric ozone has a strong seasonal cycle linked to photochemistry, with a minimum in November that increases by 30% to the seasonal maximum in June (Cooper et al., 2014). The deepest negative anomaly in 2020 occurred in July when summertime photochemical ozone production is strong, which adds weight to the argument that the 2020 anomaly is driven by a reduction in ozone precursor emissions rather than dynamics (Steinbrecht et al., 2021).

## 5. Conclusions

This paper developed a statistical framework to better quantify regional scale ozone anomalies throughout the depth of the troposphere and stratosphere by combining multiple sources of vertical profile records, such as ozonesonde, lidar and commercial aircraft data. This framework takes into account the vertical correlation structure to identify the systematic ozone variability, as well as the sampling frequency and inherent data uncertainty to determine the contribution of each data source to the fused product. Thus the regional anomalies, and their associated estimation uncertainty, can be consistently and systematically quantified.

An important implication from our finding is that regional trend assessments based on a single data source may be less reliable due to uncertainties associated with limited data, whereas incorporation of all available data sources yields more robust results. This implication is successfully demonstrated from the data integration of all ozonesonde and IAGOS records in Europe. The results are not only suitable for anomaly quantification, but also reveal fine scale ozone interannual variability, which can be useful for the evaluation of chemistry-climate

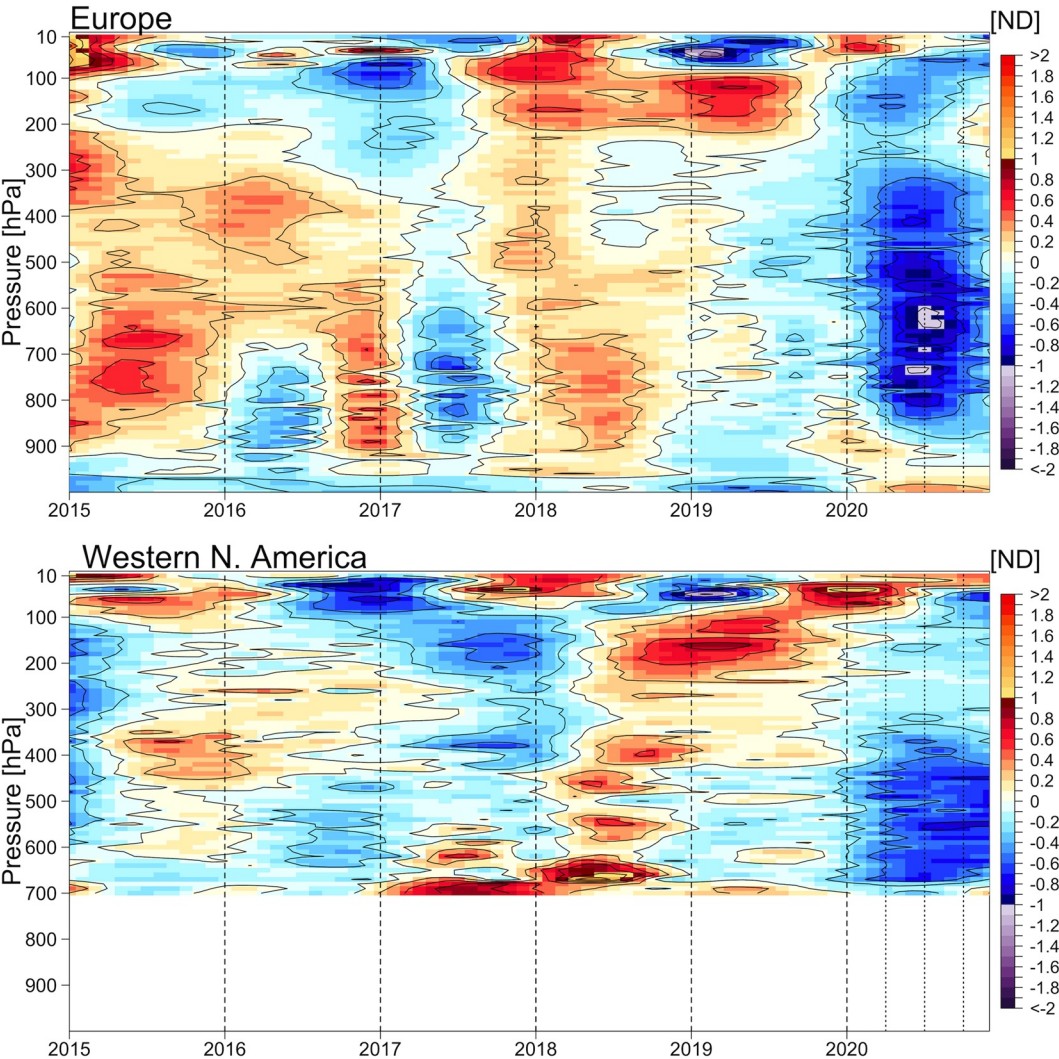

**Figure 10.** Detailed inspection of 2020 ozone anomalies above Europe and western North America, limited to the period of 2015–2020.

models. Our success producing regional ozone fields with high vertical and temporal resolution relies on abundant data samples from the IAGOS program and multiple ozonesonde stations which provide two or three profiles every week, such as Uccle, Payerne and HPB. This success also makes Europe a unique region; in contrast, data availability in western North America is less sufficient.

Our findings on the COVID-19 impact on free tropospheric ozone can be summarized as follows:

1. The directions of the long-term (1994–2019) trend estimate from individual stations across Europe and western North America are diverse, but most stations show diminishing trends when 2020 data are included, presumably due to the COVID-19 economic downturn
2. No substantial change is found above the Oceania sites (Broadmeadows, Lauder and Macquarie Island) and Hong Kong, indicating no obvious impact from the COVID-19 economic downturn. The trends in Tateno, Izaña and Hilo show a certain amount of downward changes, but are less remarkable than those at higher latitudes
3. The fused result of European IAGOS data and ozonesonde records from Uccle, Payerne, HPB, De Bilt and OHP shows that the long-term, mid-troposphere (700-300 hPa) trend decreases from 0.65 ($\pm$0.19, $p < 0.01$) ppbv/decade (1994–2019) to 0.36 ($\pm$0.20, $p < 0.01$) ppbv/decade (1994–2020) with a relative change of $-44\%$, and the quantified 2020 mean anomaly is $-3.60$ ($\pm$1.75) ppbv

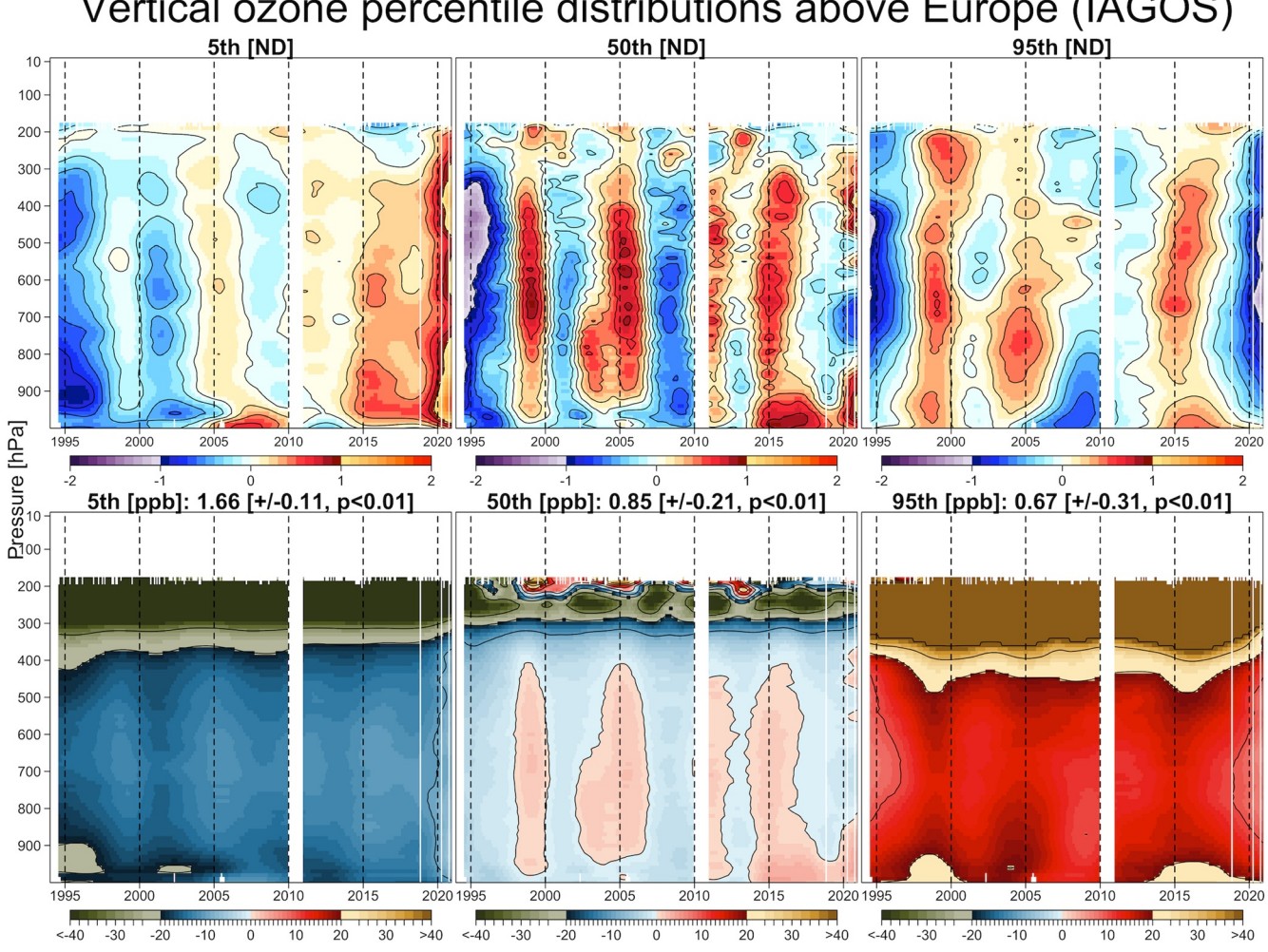

**Figure 11.** Tropospheric ozone distributions based on different percentiles above Europe. Free tropospheric (700-300 hPa) ozone trends (ppb/decade) are reported for each percentile (In-Service Aircraft for a Global Observing System data set, 1994–2020).

4. The fused result of ozonesonde, lidar and commercial aircraft data in western North America shows that the long-term, mid-troposphere trend has a relative change of −61% from 0.35 (±0.21, $p < 0.01$) ppbv/decade (1994–2019) to 0.14 (±0.21, $p = 0.19$) ppbv/decade (1994–2020), and the quantified 2020 mean anomaly is −2.77 (±1.92) ppbv

Precursor emissions and ozone levels in 2020 were anomalous and it is too early to know if emissions will return to their pre-pandemic levels (Kondragunta et al., 2021). Therefore we do not know if the positive ozone trends in the free troposphere above Europe and western North America since 1994 have stopped or if they will resume in the coming years. Continuous monitoring of free tropospheric ozone in 2021 and beyond is required to evaluate the impact of 2020 on long-term ozone trends.

Even though the vertical tropospheric ozone distributions derived in this study are based on the monthly means, the same type of analysis can be based on monthly percentiles as well. Figure 11 provides an example using the monthly fifth, 50th and 95th percentiles from the IAGOS data set above western Europe. The free tropospheric trend of the fifth percentile (1.66 (±0.11, $p < 0.01$) ppbv/decade) was twice as strong as the trend of the 95th percentile (0.67 (±0.31, $p < 0.01$) ppbv/decade) over 1994–2020. This example demonstrates that the ozone variability remains large (with respect to mean ozone trends and distribution), and therefore the mean or median distribution should not be over-interpreted as representing the most extreme level of interannual variation. This

demonstration is another example of the merits associated with abundant data sampling, since this estimation cannot be made convincingly based on sparsely sampled, once-per-week ozonesonde profiles.

## Conflict of Interest

The authors declare no conflicts of interest relevant to this study.

## Data Availability Statement

Ozonesonde data measured at Boulder, Colorado, Trinidad Head, California and Hilo, Hawaii can be accessed through ftp://aftp.cmdl.noaa.gov/data/ozwv/Ozonesonde/. The lidar records operated at Table Mountain Facility, California are available at https://lidar.jpl.nasa.gov/or ftp://ftp.cpc.ncep.noaa.gov/ndacc/RD/tmo/hdf/lidar/. Ozonesonde data measured at Hohenpeissenberg, Lindenberg (Germany), Madrid (Spain), Legionowo (Poland), Hong Kong (China), Tateno (Japan), Macquarie Island, Broadmeadows (Australia), Lauder (New Zealand), Edmonton, Kelowna and Port Hardy (Canada) can be downloaded through https://woudc.org/archive/Archive-New-Format/OzoneSonde_1.0_1/. Ozonesonde data measured at Haute-Provence Observatory (France), Izaña (Spain) and Payerne (Switzerland) can be accessed through ftp://ftp.cpc.ncep.noaa.gov/ndacc/station/. The In-Service Aircraft for a Global Observing System data are publicly available at https://doi.org/10.25326/20. The monthly quasi-biennial oscillation values can be found at https://www.geo.fu-berlin.de/met/ag/strat/produkte/qbo/qbo.dat. The monthly El Niño-Southern Oscillation index can be found at (https://psl.noaa.gov/enso/mei/).

**Acknowledgments**
This work was supported in part by the NOAA Cooperative Agreement with CIRES, NA17OAR4320101. Part of this research was carried out at the Jet Propulsion Laboratory, California Institute of Technology under a contract with the National Aeronautics and Space Administration (80NM0018D0004).

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
