## [Peer Review History · Agu Advances]

Impact of the COVID-19 economic downturn on tropospheric ozone trends: an uncertainty weighted data synthesis for quantifying regional anomalies above western North America and Europe

Kai-Lan Chang^{1,2}, Owen R. Cooper^{1,2}, Audrey Gaudel^{1,2}, Marc Allaart³, Gerard Ancellet⁴, Hannah Clark⁵, Sophie Godin-Beekmann⁴, Thierry Leblanc⁶, Roeland Van Malderen⁷, Philippe Nédélec⁸, Irina Petropavlovskikh^{1,9}, Wolfgang Steinbrecht¹⁰, René Stübi¹¹, David W. Tarasick¹², Carlos Torres¹³

¹ Cooperative Institute for Research in Environmental Sciences, University of Colorado Boulder, CO, USA

² NOAA Chemical Sciences Laboratory, Boulder, CO, USA

³ Royal Netherlands Meteorological Institute, De Bilt, The Netherlands

⁴ LATMOS, Sorbonne Université-UVSQ-CNRS/INSU, Paris, France

⁵ IAGOS-AISBL, 98 Rue du Trône, Brussels, Belgium

⁶ Jet Propulsion Laboratory, California Institute of Technology, Wrightwood, CA, USA

⁷ Royal Meteorological Institute of Belgium, Uccle, Belgium

⁸ Laboratoire d'Aérodologie, CNRS and Université de Toulouse III, Toulouse, France

⁹ NOAA Global Monitoring Laboratory, Boulder, CO, USA

¹⁰ Deutscher Wetterdienst, Hohenpeißenberg, Germany

¹¹ Federal Office of Meteorology and Climatology, MeteoSwiss, Payerne, Switzerland

¹² Environment and Climate Change Canada, Toronto, ONT, Canada

¹³ Izaña Atmospheric Research Center, AEMET, Tenerife, Spain

Files Uploaded Separately

Original Version of Manuscript (2021AV000542)

First Revision of Manuscript [Accepted] (2021AV000542R)

Author Response to Peer Review Comments

Peer Review Comments on 2021AV000542

Reviewer #1

The paper carries on the work started in Steinbrecht et al. (2021). There is no fundamental problem with this, but it is a problem, when I cannot understand where the Steinbrecht et al. (2021) paper ends and this paper starts. Specially in Section 2.2. If the results are adding more information from the first paper, it should be described more coherently. Right now there is no way to understand, why you keep referencing that it is built on the older work (what was in it that needed updating or addition) without going through that paper. That is inconvenient. Each paper should completely stand on its own, even when if it starts off from another published work

I would like to commend the author on Section 3.1. It does a good job of introducing and describing the statistical method used.

Tropospheric ozone has strong spatial signal. In this work the spatial variability has to be given up, to get a data with more frequent temporal variability. The paper should add literature on the spatial variability in the regions they are considering and comment on what they are losing out when they let go of this information

The framework of creating the fused ozone product and using it over western Europe and North America is very well discussed. But the results from this is not discussed satisfactorily. In Section 4.3, % change in various regions, various time spans, have been reported. It doesn't include any discussion about the numbers, what they mean other than some numerical values.

The title of the paper is a little misleading, given the results and discussion particularly shies away from commenting either way on the impact of covid19 related economic downturn and its impact on ozone (other than comment 1 in Discussion section). The paper itself says "Since 2020 is an anomalous year, it is still too early to know if the positive trends in the free troposphere above Europe and western North America since 1994 have stopped.."

Line 84: Tropospheric ozone reduction has always been a balance of NO_x and VOC concentration and many past studies have already established that reduction/control just one of these have not always been effective in ozone control. The authors have indicated in this sentence that although NO_x reduced, in many places ozone didn't reduce. A better description of the complete phenomenon should be discussed here, otherwise NO_x seems to be the only thing responsible for tropospheric ozone, which is misleading
Line 358: Table 1 and Table 2 has information about stations, that are not continuous from 1994-2020. Are you using the data only when it is available continuously. As I go through the paper, some of these becomes more clear, but at this point the description is unclear

Line 452: Add the relevant mathematics in Supplementary section.

Line 483: As you can see the confusion due to variable sampling duration arises earlier, while reading the paper. This should be addressed much earlier in the manuscript (preferably in Section 2.1) in more detail. You can point out that it is addressed in more detail in later sections (Section 4..)

Line 589: What is the scientific basis of this statement. Show a statistical comparison if you want make this statement, otherwise this is just a qualitative assertion

Reviewer #2

Review comments on "Impact of the COVID-19 economic downturn on tropospheric ozone trends: an uncertainty weighted data synthesis for quantifying regional anomalies above western North America and Europe" by Change et al.

This manuscript investigated the long-term trends in tropospheric ozone and identified the COVID-19 impacts over Europe and western North America. The authors developed a novel detection method based on GAMM to generate a regional-scale monthly climatology of ozone profiles and estimate tropospheric ozone anomalies. Fusing ozonesonde, lidar, and aircraft measurements, this detection framework captured the interannual variability and suggested that the year of 2020 had great negative anomalies since 1994 and diminished the long-term trends. Previous COVID-19 studies usually focus on changes in surface air quality or satellite measurements such as NO₂ column, but lack research on ozone in the free troposphere, so this manuscript provides a valuable dataset to study atmospheric chemistry. The text is concisely written and well documented. The topic is applicable for AGU Advances.

This study used an advanced statistical framework to fuse all available data and took the sampling frequency and data uncertainty into account for trend and anomaly analysis. There are two major datasets, the relatively sparse ozonesonde measurements and more frequent IAGOS aircraft data. The analysis shows that they have different values in long-term trends and anomalies (Figures 5 and S9). It could be caused by the regions where the measurements were collected (see details in the Remark below). The IAGOS data contribute significantly in the final fused data, but in 2020, the IAGOS data are very limited due to COVID-19. It makes the comparison of 1994-2019 and 1994-2020 tropospheric ozone trends difficult. I think the authors need to address this problem in the revised manuscript. details about how the data are fused are not clear in the current manuscript, and the authors need to add some explanation and/or analysis.

In summary, the current manuscript shows important results but need some further work. Minor revisions as indicated in the comments and remarks below are needed before consideration of publication in AGU Advance.

Detailed Remarks/Suggestions for Revision

Line 85: Please change 'F. Liu et al., 2020' to 'Liu et al., 2020' to keep the reference format consistent.

Line 60: A map showing the locations of these ozonesonde sites and major IAGOS airports in the supplementary material is suggested, to show how the network samples Europe and western North America.

Line 182: Same as above, please use the consistent format for (Liu et al., 2009).

Line 363: Should 'for multiple neighboring time series' be 'for multiple neighboring pressure layers'?

Line 375: How is the 'normalization' conducted? Use the multiple-year mean values as the denominator? Is the denominator calculated for each individual site or mean value at the regional scale? The authors need to add further explanation here.

Line 388: Monthly mean data are used for the analysis. Ozonesonde data are usually collected at specific time, so the monthly mean average can well represent the

tropospheric ozone. However, the IAGOS data over major cities are not always collected at the same time of day. Is it possible some of the diurnal cycle of tropospheric ozone information incorporated into the monthly mean values?

Line 389-391: Any details about the modest bias (5%) between IAGOS and ozonesonde data? Is the bias in absolute values of ozone concentrations, e.g., systematically high or low? Or the trends from these two datasets have bias?

Line 566-569: What is the cause for the 1994-1996 discrepancy between IAGOS and ozonesonde data? Different method in instrument calibration?

Line 574-575: As discussed in the paper, with and without 1994-1996 data, the trends in IAGOS and fused ozonesonde data have different response, generally decreases and increases, respectively. Therefore, when 'fuse' these two datasets together, the final fused data did not change a lot. The ozonesonde data are usually collected over rural areas (e.g., THD, TMF) or relatively small cities (e.g., Port Hardy, Hohenpeissenberg), while the IAGOS data are measured over major cities (e.g., Frankfurt, Paris, Munich, Log Angeles, San Francisco) with substantial amount of anthropogenic emissions. I am wondering if the distinct environment of atmospheric chemistry (maybe in different ozone production regime) can lead to the different responses to removal of 1994-1996 data. The revised manuscript should include further research and/or add explanation here.

Reviewer #3

Chang et al. proposes a novel and advanced statistical framework to quantify tropospheric ozone anomalies and trends in EU and western US by combining vertical ozone profiles measurements from ozonesonde and IAGOS data. The method is a further step based on their previous work (Chang et al., GMD, 2020). They then use this method to identify significant ozone negative anomaly in 2020 due to the COVID 2019. Overall the method is elegant, comprehensively described and discussed. The paper is well written though sometimes a bit technical. The result is convincing. I recommend publication after some minor concerns are addressed.

1. One difficulty in analyzing data from IAGOS or ozonesonde for tropospheric ozone trend studies may be how to exclude ozone from stratosphere. I notice that the authors removed ozone from stratospheric air mass with high PV values in their previous ozone trend studies (Gaudel et al., 2020; Chang et al., 2020), but there is no description in this work (or I just missed it). I am actually not sure whether the authors should consider remove stratospheric ozone influences in tropospheric ozone trend studies. It looks to me that if the paper is focusing on anthropogenic precursor driven trends then removing stratospheric ozone influences is reasonable, but changes in STE may be a potential driver of tropospheric ozone trend, and in this case removing stratospheric ozone influences may be inappropriate. I would expect a more insightful discussion on this issue in this work.

2. Line 250, Formula (1). It looks similar as that in Cooper et al. (2020) but with the ENSO and QBO terms added. Cooper et al. (2020) used deseasonalized monthly anomaly of

ozone as $y(t)$. Is it the same here? In later discussion (line 361) I do see deseasonalization but not sure whether it is also applied in Formula 1. Please clarify. Also, is there any process applied to the ENSO and QBO index?

3. Table 1. I think adding a map plot with sites labeled would be much helpful for non-US and non-EU scientists.

4. Table 2. Sites Uccle, HPB, and Payerne all have very high sample frequency, but they show totally different trends in the free troposphere. What is the implication? Can we infer that ozonesonde measurements are not representative for regional ozone trends?